# High-resolution quantitative and functional MRI indicate lower myelination of thin and thick stripes in human secondary visual cortex

Daniel Haenelt[1,2]*, Robert Trampel[1], Shahin Nasr[3,4], Jonathan R Polimeni[3,4,5], Roger BH Tootell[3,4], Martin I Sereno[6], Kerrin J Pine[1], Luke J Edwards[1], Saskia Helbling[1,7], Nikolaus Weiskopf[1,8]

[1]Department of Neurophysics, Max Planck Institute for Human Cognitive and Brain Sciences, Leipzig, Germany; [2]International Max Planck Research School on Neuroscience of Communication: Function, Structure, and Plasticity, Leipzig, Germany; [3]Athinoula A. Martinos Center for Biomedical Imaging, Massachusetts General Hospital, Charlestown, United States; [4]Department of Radiology, Harvard Medical School, Boston, United States; [5]Harvard-MIT Division of Health Sciences and Technology, Massachusetts Institute of Technology, Cambridge, United States; [6]Department of Psychology, College of Sciences, San Diego State University, San Diego, United States; [7]Poeppel Lab, Ernst Strüngmann Institute (ESI) for Neuroscience in Cooperation with Max Planck Society, Frankfurt am Main, Germany; [8]Felix Bloch Institute for Solid State Physics, Faculty of Physics and Earth Sciences, Leipzig University, Leipzig, Germany

*For correspondence:
haenelt@cbs.mpg.de

**Abstract** The characterization of cortical myelination is essential for the study of structure-function relationships in the human brain. However, knowledge about cortical myelination is largely based on post-mortem histology, which generally renders direct comparison to function impossible. The repeating pattern of pale-thin-pale-thick stripes of cytochrome oxidase (CO) activity in the primate secondary visual cortex (V2) is a prominent columnar system, in which histology also indicates different myelination of thin/thick versus pale stripes. We used quantitative magnetic resonance imaging (qMRI) in conjunction with functional magnetic resonance imaging (fMRI) at ultra-high field strength (7 T) to localize and study myelination of stripes in four human participants at sub-millimeter resolution in vivo. Thin and thick stripes were functionally localized by exploiting their sensitivity to color and binocular disparity, respectively. Resulting functional activation maps showed robust stripe patterns in V2 which enabled further comparison of quantitative relaxation parameters between stripe types. Thereby, we found lower longitudinal relaxation rates ($R_1$) of thin and thick stripes compared to surrounding gray matter in the order of 1–2%, indicating higher myelination of pale stripes. No consistent differences were found for effective transverse relaxation rates ($R_2^*$). The study demonstrates the feasibility to investigate structure-function relationships in living humans within one cortical area at the level of columnar systems using qMRI.

## Editor's evaluation

Using a combination of cutting-edge high-resolution magnetic resonance protocols, this important study investigates the structure-function relationship of specialised compartments in the human visual cortex in vivo. The use of quantitative MRI provides convincing evidence that color- and

disparity-selective functional compartments (thin and thick stripes) of visual area V2 have different MR relaxation properties than the pale stripes of V2. While these results indicate different patterns of myelination across the "stripes" of V2, the MR signals will require independent validation to be considered specific to myelin. This study will be of interest to a wide range of neuroscientists and clinicians employing imaging methods to study cortical organization.

## Introduction

In primates, visual information sent from the primary visual cortex (V1) to the secondary visual cortex (V2) is segregated into distinct modules known as thin, thick, and pale stripes (*Hubel and Livingstone, 1987*; *Livingstone and Hubel, 1987*). These stripes form a columnar system in the sense that their functional properties extend roughly through cortical depth (*Tootell and Hamilton, 1989*). Functional properties include the sensitivity to different visual features like color, orientation, binocular disparity, and motion, which are largely processed in different stripe types and sent to distinct cortical areas. For example, thin stripes are sensitive to color content and project to functional area V4, whereas thick stripes are more sensitive to binocular disparity and project to area MT (V5) (*Shipp and Zeki, 1985*; *Hubel and Livingstone, 1987*; *Livingstone and Hubel, 1987*; *Ts'o et al., 2001*).

Using cytochrome oxidase (CO) staining, these stripes were found first in squirrel monkeys and macaques as dark and pale patches organized in repeating pale-thin-pale-thick cycles, running through V2 and oriented approximately perpendicular to the V1/V2 border (*Livingstone and Hubel, 1982*; *Tootell et al., 1983*). In macaques, stripes of the same type have a center-to-center distance of around 4.0 mm and a width ranging from 0.7 to 1.3 mm (*Shipp and Zeki, 1985*; *Tootell and Hamilton, 1989*). In humans, these widths are approximately doubled in size (*Hockfield et al., 1990*; *Tootell and Taylor, 1995*; *Adams et al., 2007*).

Histological studies also showed a stripe pattern in V2 of post-mortem brain specimens when techniques for the staining of myelin were used (*Tootell et al., 1983*; *Krubitzer and Kaas, 1989*; *Horton and Hocking, 1997*). However, these studies gave an inconsistent picture of the correspondence between stripes defined by CO activity and myelin density. Staining with Luxol fast blue indicated stronger myelination in thin/thick (*Tootell et al., 1983*) stripes, while Gallyas silver staining showed pale (*Krubitzer and Kaas, 1989*) stripes being more myelinated. This discrepancy between myelin staining methods was replicated in another study in which several methods were compared to each other (*Horton and Hocking, 1997*). In addition to inconsistencies across staining methods, all standard histochemical methods are highly sensitive to the condition of the brain specimen (e.g. post-mortem delay time), variations in fixation and staining procedures, and exposure time (*Savaskan et al., 2009*).

Magnetic resonance imaging (MRI) provides an alternative view to study tissue microstructure in living humans with the possibility to generate a multitude of image contrasts which can specifically be sensitized to myelin (*Edwards et al., 2018*; *Weiskopf et al., 2021*). Quantitative MRI (qMRI) is a collective term of techniques that aims to isolate one source of image contrast and represent it as a single quantitative parameter map, for example, maps of longitudinal relaxation rate ($R_1$), effective transverse relaxation rate ($R_2$*), proton density ($PD$), or magnetic susceptibility. Parameter maps are created by combining multiple 'weighted' MR images in a model-based fashion yielding reproducible and standardized measures in physical units, which are less dependent on the acquisition (*Edwards et al., 2018*; *Trampel et al., 2019*; *Weiskopf et al., 2021*). Furthermore, by separating different sources of image contrast, qMRI provides a closer relationship to specific microstructural components, for example, myelin and iron (*Edwards et al., 2018*; *Weiskopf et al., 2021*). In this regard, it should be kept in mind that single qMRI parameter maps do not have the specificity to directly infer the abundance of single microstructural tissue components by being sensitive to multiple tissue components to a varying extent and therefore can only serve as a surrogate, for example, in the study of cortical myelination (*Lazari and Lipp, 2021*). Nevertheless, concerning $R_1$, it is known that cortical $R_1$ is largely affected by myelin content (*Stüber et al., 2014*; *Leuze et al., 2017*) and, hence, was often used as surrogate to study the macroscopic distribution of cortical myelination (*Sigalovsky et al., 2006*; *Dick et al., 2012*; *Sereno et al., 2013*; *Lutti et al., 2014*). In addition, it might be possible to exploit the complementary multi-modal information from different

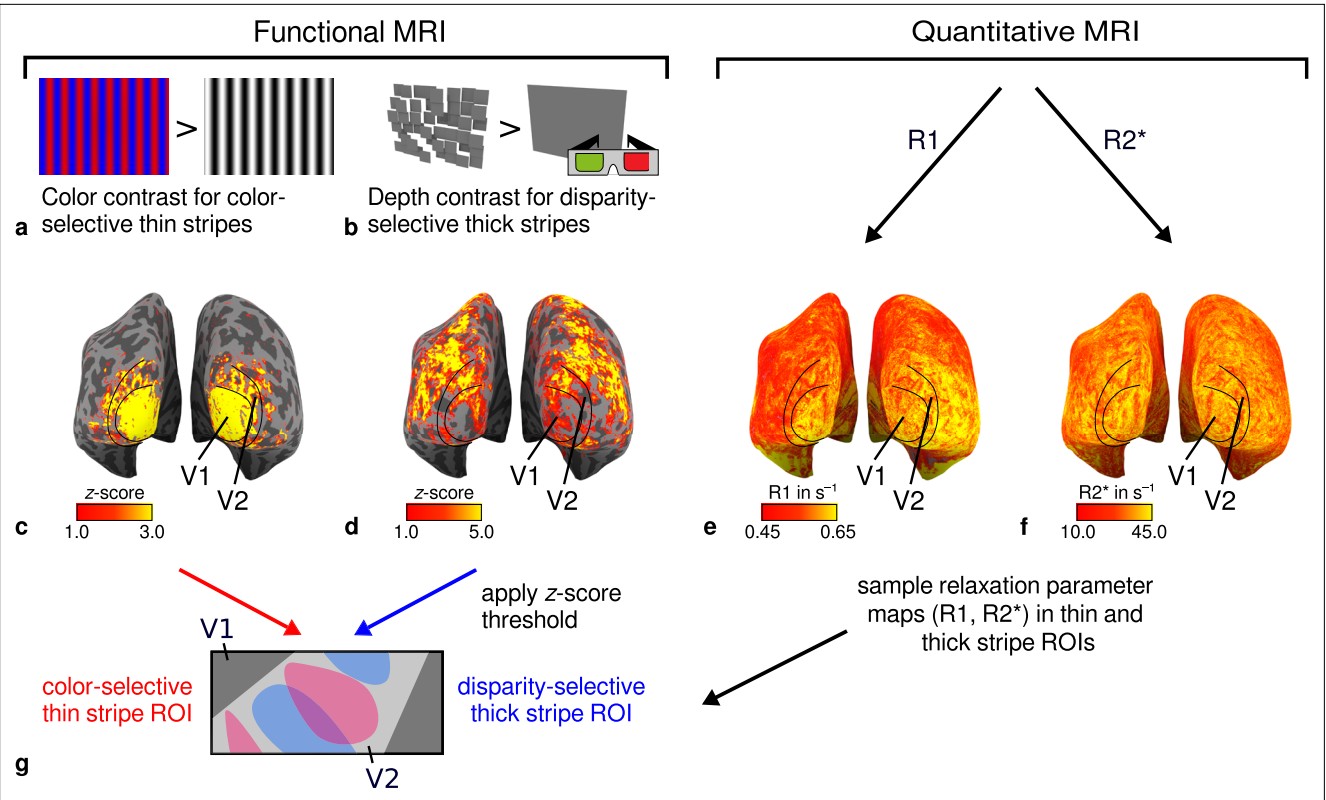

**Figure 1.** General overview of acquired magnetic resonance (MR) data and their use in the analysis. (**a**) Example of chromatic and achromatic stimuli used to map color-selective thin stripes. (**b**) Schematic illustration of stimuli when viewed through anaglyph spectacles used for mapping disparity-selective thick stripes. These stimuli consisted of a disparity-defined checkerboard and a plane intersecting at zero depth, respectively. Exemplary activation maps from thin stripe (contrast: color > luminance) and thick stripe (contrast: depth > no depth) mapping sessions are shown for a representative participant (subject 3) in (**c**) and (**d**), respectively. Quantitative $R_1$ and $R_2^*$ maps from the same participant are shown in (**e**–**f**). (**g**) Activation maps from (**c**) and (**d**) were used to define regions of interest (ROIs) for thin- and thick-type stripes in V2 by applying a z-score threshold. $R_1$ and $R_2^*$ values were sampled in these ROIs for further analysis. Borders in (**c**–**f**) were manually defined on the basis of a separate retinotopy measurement.

qMRI parameter maps using appropriate biophysical models to directly quantify the underlying tissue microstructure with higher specificity, which would open the way to in vivo histology (*Weiskopf et al., 2021*).

Functional MRI (fMRI) allows in vivo localization of functional architecture. Recent developments in ultra-high field MRI enabled the functional localization of thin and thick stripes using high-resolution fMRI (*Nasr et al., 2016*; *Dumoulin et al., 2017*; *Navarro et al., 2021*) by, for example, exploiting their different sensitivity to color (*Tootell et al., 1983*; *Tootell et al., 2004*) and binocular disparity (*Peterhans and von der Heydt, 1993*; *Chen et al., 2008*), respectively (*Nasr et al., 2016*). This allows MR-based investigations of mesoscale structure-function relationships in the same living participant, that is, at the spatial scale of cortical columns and layers.

We combined the localization of V2 stripes using high-resolution fMRI with qMRI measurements to infer myelination differences between stripe types. We robustly show lower $R_1$ values in color-selective thin and disparity-selective thick stripes in comparison to locations containing pale stripe contributions, which points toward higher myelin density in pale stripes. Whereas recent studies have explored cortical myelination in V2 in macaques (*Li et al., 2019*) and humans (*Dumoulin et al., 2017*) using non-quantitative, weighted MR images, to the best of our knowledge, we showed for the first time myelination differences using MRI on a quantitative basis at the spatial scale of columnar systems. This shows the feasibility to use high-resolution qMRI in conjunction with high-resolution fMRI to study the relationship between functional and structural properties of the brain in living humans, which is a fundamental goal in neuroscience.

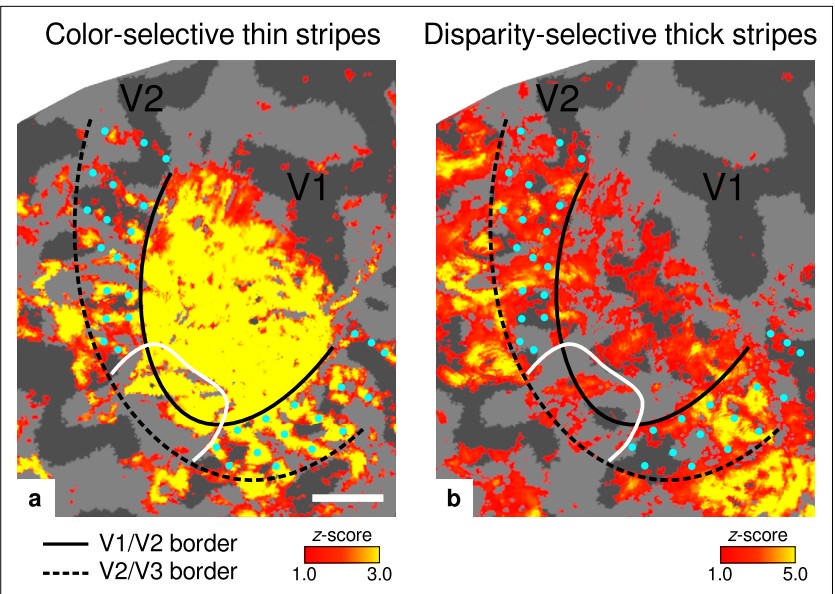

**Figure 2.** Activation maps for color-selective thin and disparity-selective thick stripes. Thin stripes (contrast: color > luminance) and thick stripes (contrast: depth > no depth) are shown as thresholded activation maps in (**a**) and (**b**), respectively. Both maps were averaged across sessions, sampled at mid-cortical depth, and are illustrated on the flattened surface of the right hemisphere for one representative participant (subject 3). Surfaces were flattened using FreeSurfer (6.0.0, http://surfer.nmr.mgh.harvard.edu/) after cutting out a region on the surface mesh which included all stimulated portions of V1 and V2. Data from all participants can be found in Appendix 1. In V2, patchy stripes can be identified, which run through V2 oriented perpendicular to the V1/V2 border. Borders were manually defined on the basis of a separate retinotopy measurement, which is illustrated in *Figure 2—figure supplement 1* (see Appendix 1 for retinotopy data from all participants). The white border illustrates a region around the foveal representation with missing activation. This region was excluded in the supporting analysis shown in *Figure 5—figure supplement 2*. For the main analysis, all data points in V2 were used. Manually drawn cyan dots mark activated regions in (**a**) to illustrate the alternating activation pattern between (**a**) and (**b**). Cortical curvature is shown in gray (sulcal cortex is dark gray and gyral cortex light gray). Another example can be found in *Figure 2—figure supplement 2*. Scale bar: 1 cm.

The online version of this article includes the following figure supplement(s) for figure 2:

**Figure supplement 1.** Eccentricity and polar angle maps.

**Figure supplement 2.** Exemplary activation maps from another participant.

## Results

Participants (*n* = 4) were invited for multiple fMRI and qMRI sessions at 7 T (see *Figure 1*). On different days, we measured high-resolution (0.8 mm isotropic) fMRI responses to stimuli varying in color and binocular disparity content, respectively, to locate color-selective thin stripes (color stripes) and disparity-selective thick stripes (disparity stripes) in V2 (*Nasr et al., 2016*). We use the terms color-selective thin and disparity-selective thick stripes acknowledging the close relationship between color processing and CO thin stripes (*Tootell et al., 2004*) and disparity processing and CO thick stripes (*Chen et al., 2008*), respectively, despite not having directly measured CO content in this study. In a separate session, we used the multi-parameter mapping (MPM) protocol (*Vaculčiaková et al., 2022*) to acquire high-resolution anatomical images with 0.5 mm isotropic resolution from which quantitative parameter maps ($R_1$, $R_2$*, *PD*) were derived.

### Functional mapping of color-selective and disparity-selective stripes

Color- and disparity-selective stripes were identified in each individual in separate scanning sessions. *Figure 2* shows activation maps averaged over two sessions and sampled at mid-cortical depth from one representative participant (see Appendix 1 for activation maps from all participants).

Color-selective thin stripes can be identified in *Figure 2a* with expected topography (*Tootell et al., 1983*; *Nasr et al., 2016*), that is, they start at the V1/V2 border, radiate outward in parallel, and are

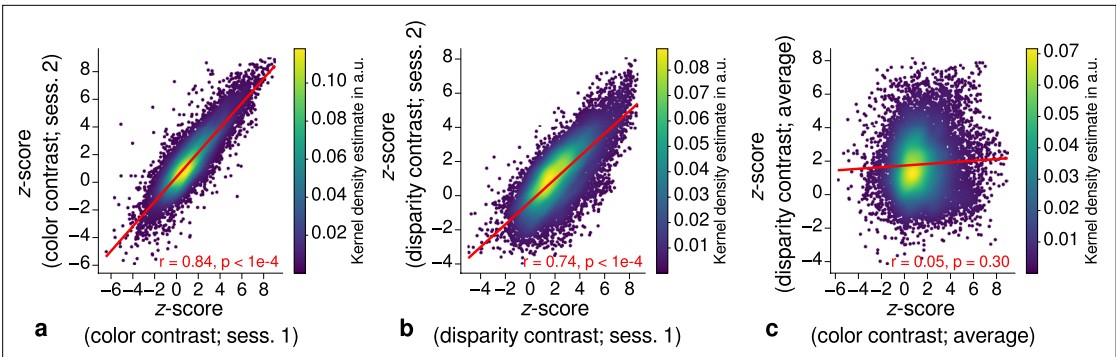

**Figure 3.** Repeatability of functional magnetic resonance imaging (fMRI) activation maps across scanning sessions. Scatter plots with kernel density estimation illustrate the consistency of activation maps across scanning sessions for one representative participant (subject 3). Sessions were carried out on different days and activation maps were sampled at mid-cortical depth. (**a**) shows correspondences of $z$-scores in the secondary visual cortex (V2) between single color-selective thin stripe mapping sessions (contrast: color > luminance). (**b**) shows the same for single disparity-selective thick stripe mapping sessions (contrast: depth > no depth). In (**c**), correspondences of average $z$-scores (across sessions) between thin and thick stripe sessions are shown. Regression lines are indicated as red lines. Spearman's rank correlation coefficients $r$ and $p$-values determined by permutation analysis (see Materials and methods) are annotated inside the plots and demonstrate high repeatability of color-selective thin and disparity-selective thick stripe scanning sessions. Note that the comparison between thin and thick stripe sessions shows no statistically significant correlation as expected from the interdigitated nature of both stripe types. Plots for all participants can be found in *Figure 3—figure supplement 1* and *Figure 3—figure supplement 2*.

The online version of this article includes the following figure supplement(s) for figure 3:

**Figure supplement 1.** Correlation plots for single participants (subjects 1–2).

**Figure supplement 2.** Correlation plots for single participants (subjects 3–4).

confined to area V2. *Figure 2b* shows locations selective for binocular disparity. Activation maps for binocular disparity showed a less pronounced stripe pattern in V2. It should be noted that color-selective stripes are known to be largely confined to CO thin stripes (*Xiao et al., 2003*; *Tootell et al., 2004*), whereas selectivity for binocular disparity is found in all stripe types but most frequently in CO thick stripes (*Peterhans and von der Heydt, 1993*; *Chen et al., 2008*). In V1, no activation was found for binocular disparity in *Figure 2b* which is consistent with findings by *Tsao et al., 2003*; *Nasr et al., 2016*, while large V1 activation was found for color contrast in *Figure 2a* as also shown by *Nasr et al., 2016*.

Cyan dots were added in *Figure 2* to qualitatively illustrate the alternation of activation clusters between stripe types as expected from the thin/thick stripe arrangement. We note that, as *Figure 2a* and *Figure 2b* show results from two independent experiments, the alternating stripe pattern is not an intrinsic outcome of the experimental design.

Each stripe type was localized in two independent scanning sessions and activation maps were consistent between sessions of color and disparity stripe measurements, respectively. This is illustrated in *Figure 3*, which shows statistically significant correlations of activation maps between sessions for one representative participant.

## Consistent qMRI maps across cortical regions and cortical depth

*Figure 4a–b* shows an $R_1$ map sampled at mid-cortical depth for one representative participant. Primary motor and primary sensory cortical areas have higher $R_1$ values, congruent with higher myelin density in these areas (*Flechsig, 1920*; *Glasser and Van Essen, 2011*; *Sereno et al., 2013*). To further check the consistency of our data with literature, we qualitatively compared cortical mean $R_1$ parameters between several cortical regions of interest (ROIs) with known myelination differences. ROIs were defined by probabilistic FreeSurfer (6.0.0, http://surfer.nmr.mgh.harvard.edu/) labels for each participant. First, we used the FreeSurfer Brodmann area maps of V1, V2, and MT (V1_exvivo.thresh.label, V2_exvivo.thresh.label and MT_exvivo.thresh.label) (*Fischl et al., 2008*; *Hinds et al., 2008*). Second, we defined an angular gyrus label from the FreeSurfer parcellation (*Destrieux et al., 2010*).

*Figure 4c* shows systematic $R_1$ variations with highest values in V1 for each participant, which is in line with Fig. 1(b) in *Sereno et al., 2013*. *Figure 4—figure supplement 1* and *Figure 4—figure*

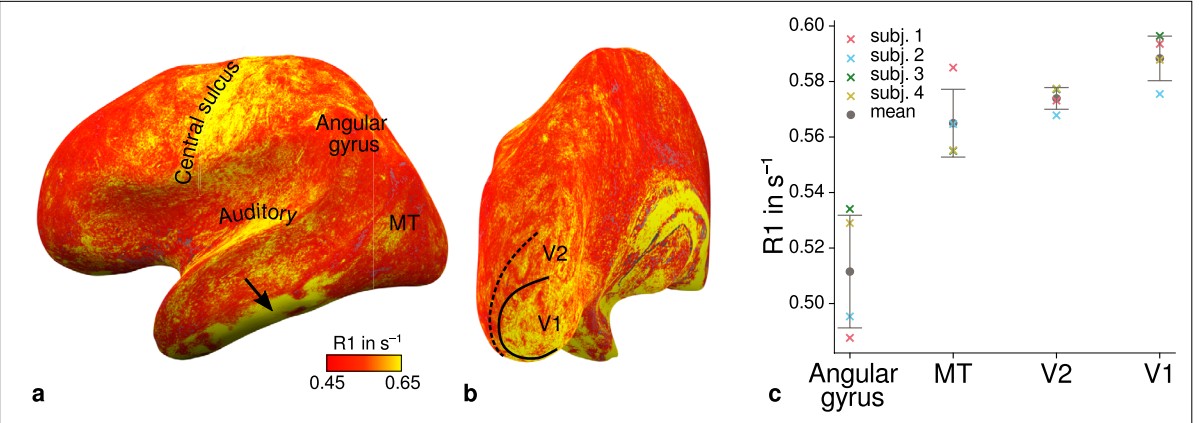

**Figure 4.** Illustration of quantitative $R_1$ across cortical areas. Cortical $R_1$ values are shown at mid-cortical depth of the left hemisphere on an inflated surface from a representative participant (subject 3) in lateral (**a**) and posterior (**b**) view. Higher $R_1$ values can be qualitatively identified in primary motor and sensory areas, which reflects known cortical myeloarchitecture (***Flechsig, 1920***; ***Glasser and Van Essen, 2011***). The arrow in (**a**) points to an artifact outside of V2 caused by magnetic field inhomogeneities. Data from all participants can be found in Appendix 1. (**c**) Mean $R_1$ values are shown for different cortical regions (angular gyrus, MT, V2, V1) defined by corresponding FreeSurfer labels (***Fischl et al., 2008***; ***Hinds et al., 2008***; ***Destrieux et al., 2010***) of each participant (similar to Fig. 1(b) in ***Sereno et al., 2013***). All participants show increased $R_1$ values in V1. Corresponding source data is given in ***Figure 4—source data 1***. A corresponding illustration for quantitative $R_2^*$ can be found in ***Figure 4—figure supplement 1***. ***Figure 4— figure supplement 2*** shows mean $R_1$ (based on a separate whole-brain MP2RAGE acquisition) and $PD$ values across cortical areas. Higher $R_1$ values in V1 as shown in (**c**) could be confirmed with the independent $R_1$ estimate from the MP2RAGE acquisition. Mean across participants is shown in gray. Vertical error bars indicate 1 standard deviation across participants.

The online version of this article includes the following source data and figure supplement(s) for figure 4:

**Figure supplement 1.** Illustration of quantitative $R_2^*$ across cortical areas.

**Figure supplement 2.** Mean quantitative $R_1$ (MP2RAGE) and $PD$ values across cortical areas.

**Source data 1.** Source data for mean quantitative magnetic resonance imaging (qMRI) values ($R_1$, $R_2^*$, $PD$) across cortical areas (angular gyrus, MT, V2, V1).

supplement 2 illustrate the same comparison for $R_2^*$ and $PD$ values. Whereas $R_2^*$ values showed similar results, $PD$ lacked a consistent trend across participants. This might be due to remaining receiver bias in final $PD$ maps, which is challenging to remove especially at high magnetic field strengths (***Volz et al., 2012***). We therefore did not consider $PD$ parameter maps for the main analysis. We also checked cortical profiles of mean parameters in V2 by sampling data on surfaces defined at different cortical depths (see Appendix 2). In all participants, we confirmed the expected decrease of $R_1$, $R_2^*$, and $MTVF = 100\% - PD$ (macromolecular tissue volume fraction [***Mezer et al., 2013***]) values toward the pial surface since all three parameters are sensitive to myelin (***Marques et al., 2017***; ***Carey et al., 2018***; ***Kirilina et al., 2020***).

## Higher myelination of pale stripes

Using $R_1$ and $R_2^*$ as surrogates for cortical myelination, we tested whether different stripe types are differentially myelinated by comparing $R_1$ and $R_2^*$ parameter values between stripe types following a similar procedure described in ***Li et al., 2019***. In brief, color-selective thin and disparity-selective thick stripe ROIs were demarcated by applying a $z$-score threshold to the corresponding functional contrasts (see Appendix 3 for a depiction of ROIs for each participant defined at $z = 1.96$). Mean $R_1$ and $R_2^*$ from one stripe type were then tested against the mean value within V2 excluding data belonging to the other stripe type (see Materials and methods). This enabled us to indirectly demarcate pale stripes assuming a strict tripartite stripe division of V2. Since the definition of ROIs solely based on $z$-score thresholds is inevitably arbitrary, we performed the above analysis for several thresholds. ***Figure 5*** shows the pooled $R_1$ and $R_2^*$ for $z \in \{0, 0.5, \ldots, 4.5\}$ across participants. Quantitative parameter values are shown as deviation from the mean within V2 after regressing out variations due to local curvature. For each $z$-score threshold level, we tested the difference for statistical significance using permutation testing. ***Figure 5a–b*** shows statistically significant differences of $R_1$ between thin or thick stripes and mean of V2 excluding the other stripe type, which points toward higher myelin

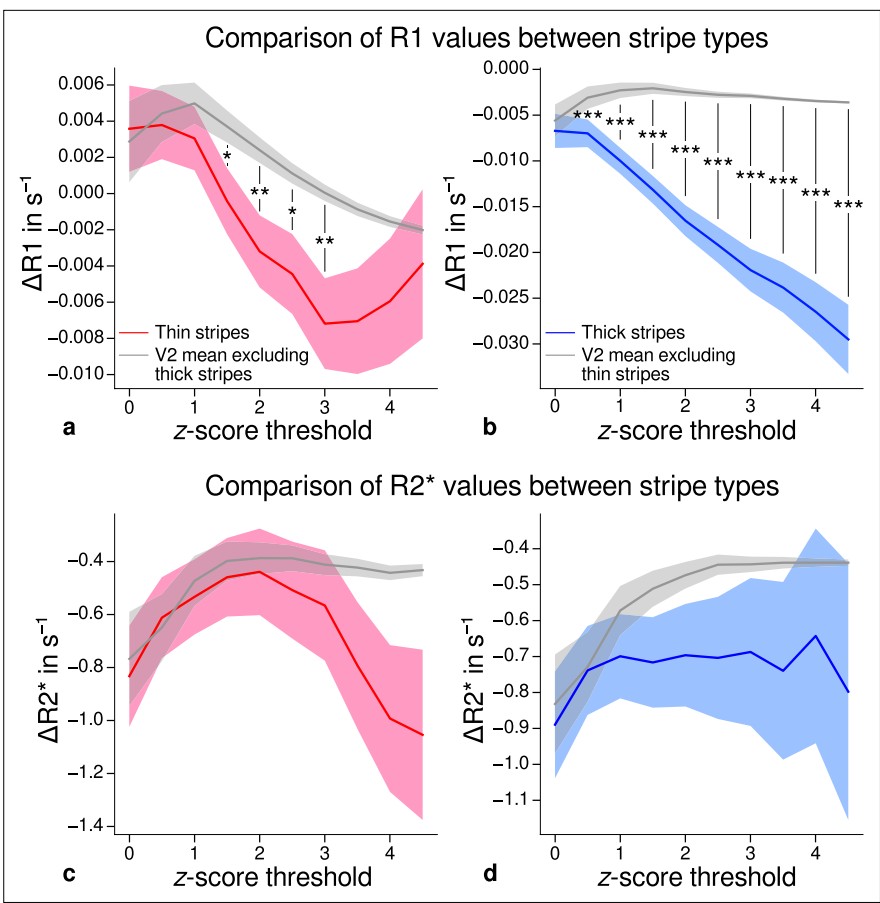

**Figure 5.** Comparison of quantitative $R1$ and $R_2^*$ values between V2 stripe types. Cortical $R_1$ (**a–b**) and $R_2^*$ (**c–d**) values in thin stripes (red), thick stripes (blue), and whole V2 excluding the other stripe type (gray; and therefore containing contributions from pale stripes) are shown for various $z$-score threshold levels, which were used to define thin and thick stripe regions of interest (ROIs). Quantitative values are illustrated as deviation from the mean within V2 after removing variance from local curvature. Values were pooled across participants and hemispheres. Differences between data in thin/thick stripes and whole V2 without thick/thin stripes were tested for statistical significance at $z \in \{0, 0.5, \ldots, 4.5\}$. Statistical significance was assessed by permutation testing (see Materials and methods). $R_1$ in both thin and thick stripes is lower than surrounding gray matter, which points toward higher myelination of pale stripes assuming a strict tripartite stripe division in human V2. No effects were found for $R_2^*$. A supporting analysis confirmed these results when excluding the region around the foveal representation from the analysis. This addresses a potential bias due to missing activation around the foveal representation in functional activation maps, which were used to define stripe ROIs. This is shown in *Figure 5—figure supplement 1*. The results for $R_1$ values were further confirmed using an independent estimate of cortical $R_1$ based on separately acquired whole-brain MP2RAGE scans, which is shown in *Figure 5—figure supplement 2* and *Figure 5—figure supplement 3*. Corresponding source data is given in *Figure 5—source data 1*. Statistically significant differences are marked by asterisks, *: p < 0.05, **: p < 0.01, ***: p < 0.001. Shaded areas indicate 1 standard deviation of the generated null distribution used for permutation testing.

The online version of this article includes the following source data and figure supplement(s) for figure 5:

**Figure supplement 1.** Comparison of quantitative $R_1$ and $R_2^*$ values between V2 stripe types excluding foveal parts.

**Figure supplement 2.** Comparison of quantitative $R_1$ values (MP2RAGE) between V2 stripe types.

**Figure supplement 3.** Comparison of quantitative $R_1$ values (MP2RAGE) between V2 stripe types excluding foveal parts.

**Source data 1.** Source data for comparisons of $R_1$ and $R_2^*$ between V2 stripe types.

density in pale stripes. These results were confirmed by an independent data set using $R_1$ values estimated from the MP2RAGE sequence (*Marques et al., 2010*), which is shown in *Figure 5—figure supplement 2*. The maximum $z$-score threshold was chosen arbitrarily and was limited by the resulting ROI size. ROI sizes for all threshold levels and participants are illustrated in Appendix 4. Note that higher thresholds lead to an expansion of the pale stripe ROI and contamination from other stripe types. In *Figure 5*, shaded areas denote the standard deviation of the generated null distribution used for permutation testing. This illustrates the enlargement of pale stripe ROIs at high threshold levels since larger ROIs lead to less variation across permutations. For an intermediate threshold level of $z = 1.96$ (p < 0.05, two-sided), $R_1$ values in thin and thick stripes differ from pale stripes by 0.005 s$^{-1}$ and 0.014 s$^{-1}$, respectively, which corresponds to a deviation of around 1–2% assuming a mean longitudinal relaxation rate of 0.58 s$^{-1}$ in V2 (see *Figure 4c*). No statistically significant effects were found for $R_2^*$ as shown in *Figure 5c–d*.

From functional contrasts shown in *Figure 2* (see also Appendix 1 and Appendix 3), it is evident that the region around the foveal representation of V2 largely did not show functional activation. Since we used all portions of V2 in the analysis which were covered by retinotopy measurements, this potentially could have biased our results. To check if missing activation around the foveal representation affected our results, we performed an additional analysis, in which we excluded that foveal region. More precisely, we used the phase responses from retinotopy measurements (eccentricity) and constructed masks to exclude the inner one-third of the full phase cycle in individual activation maps (see white lines in *Figure 2* and Appendix 1). The excluded eccentricity range was a compromise between masking out a sufficient region around the foveal representation with no functional activation and leaving enough data points for analysis. Comparison of qMRI parameters yielded basically the same results both for MPM and for MP2RAGE data shown in *Figure 5—figure supplement 1* and *Figure 5—figure supplement 3*, respectively. Note that *Figure 5—figure supplement 1* also shows significant differences for $R_2^*$ which also points into the direction of higher myelin content of pale stripes without being as conclusive as $R_1$.

## Discussion

The secondary visual cortex of the primate contains a repeating pale-thin-pale-thick stripe pattern of CO activity. It is known that components of visual information like color, orientation, and binocular disparity are largely segregated into separate pathways and processed in different stripe types (*Hubel and Livingstone, 1987*; *Livingstone and Hubel, 1987*). We robustly mapped color-selective thin and disparity-selective thick stripes in humans using high-resolution fMRI. Furthermore, we demonstrated that $R_1$ parameter maps reflect the expected macroscopic features of cortical myelination in primary motor and sensory areas and across cortical depth. By combining data from fMRI and qMRI, we then showed that locations in V2 have higher $R_1$ values which neither correspond to functionally defined thin nor thick stripes. Because myelin content is a major contrast mechanism for cortical $R_1$ (*Stüber et al., 2014*) and we expect a tripartite stripe architecture, we interpret this as an indication of higher myelination of pale stripes than thin and thick stripes in V2.

It is important to note that $R_1$ is used as a surrogate for cortical myelination and histological findings of myelin content in V2 are not conclusive so far. On the one hand, *Tootell et al., 1983*, showed higher myelination in dark CO stripes in monkey V2 using the Luxol fast blue staining technique and using wet unstained brain tissue sections. On the other hand, *Krubitzer and Kaas, 1989* showed higher myelination in pale CO stripes using the Gallyas staining technique. These conflicting findings were reproduced in a later study by *Horton and Hocking, 1997*. This study also examined myelination in monkey V1 after monocular enucleation. After monocular enucleation, CO activity is down-regulated in layer 4c neurons located in ocular dominance columns (ODCs) of the deafferented eye. This enables the visualization of ODCs of the missing eye, which appear as light columns after CO staining. Monocular enucleation further leads to atrophy of the optic nerve and loss of myelin through the process of Wallerian degeneration. Unexpectedly, *Horton and Hocking, 1997*, showed higher myelination in ODCs of the missing eye in wet unstained tissue and using myelin basic protein immunohistochemistry. Furthermore, wet unstained sections showed higher myelination in V2 dark CO stripes of normal monkeys. Both findings would be in conflict with each other if a relationship between patches/stripes of CO activity and myelination is assumed. Both classical myelin stains (Luxol fast blue and Gallyas) showed the expected higher myelination in ODCs of the intact eye. However, it is known that Luxol

fast blue, which stained CO-rich regions in both V1 and V2, is not only sensitive to myelin content but also shows an affinity to mitochondria where CO is mostly found. This dual character might have influenced the staining results in CO-rich cortical regions of V1 and V2. The Gallyas staining showed the expected higher myelination in ODCs of the normal eye after monocular enucleation and higher myelination in V2 pale stripes. While this does not reveal a complete and conclusive picture, we reproduced the results obtained with the Gallyas staining method using qMRI. Our results were further corroborated by a recent MRI study in macaques (*Li et al., 2019*), which used a similar approach to define thin and thick stripes as in the present study.

However, our results do not align with a recent human MRI study by *Dumoulin et al., 2017*, which found higher myelin density in thick stripes using a $T_1$-weighted imaging sequence (MPRAGE) to infer myelination differences. Although the reason for this discrepancy cannot be conclusively determined, two aspects that differ between studies are worth mentioning. First, the functional localization of stripes was different. In *Dumoulin et al., 2017*, parvo- and magnocellular dominated pathways were targeted by exploiting known differences in the processing of slow and fast temporal frequencies in the visual stimulus, respectively. However, the assignment of parvo- and magnocellular streams to particular stripes in V2 is still controversial (*Sincich and Horton, 2005*). Thus, their relation to the tripartite stripe architecture is less clear than for color content and binocular disparity as used in our study (*Hubel and Livingstone, 1987*; *Tootell et al., 2004*; *Chen et al., 2008*). Second, in contrast to our study, myelin density was inferred from weighted MR images, which are known to be more affected by technical biases.

The observed differences of myelin density between stripe types were based on $R_1$ estimates using the MPM protocol in the main analysis. We further confirmed these results with an independent data set using $R_1$ estimates from the MP2RAGE acquisition (see *Figure 5—figure supplement 2*). This further demonstrates the generalizability of our results across acquisition methods. Nevertheless, although previous studies show the validity of $R_1$ as a myelin marker in this type of healthy participant group with no known pathologies (*Stüber et al., 2014*; *Leuze et al., 2017*) and differences were found with two different $R_1$ mapping methods, we cannot fully exclude that some unknown factors may systematically bias the indirect measures.

We did not find any significant differences of $R_2^*$ between stripe types in the main analysis as shown in *Figure 5c–d*. Whereas $R_1$ in the normal cortex is largely influenced by myelination levels, $R_2^*$ is sensitive to paramagnetic iron and diamagnetic myelin (*Stüber et al., 2014*; *Kirilina et al., 2020*). Other factors like vasculature and the orientation of the cortex to the static magnetic field of the MR system have an influence on $R_2^*$, which might have obscured the underlying dependency on myelin content (*Cohen-Adad et al., 2012*). The dark appearance of thin and thick stripes after CO staining is a marker for increased oxidative metabolism compared to pale stripes. This favors the hypothesis of richer vascularization in thin and thick stripes, which potentially could counteract reductions in $R_2^*$ due to lower myelination. Indeed, higher vessel densities were found in blobs (another CO-rich structure in V1) and stripes of squirrel and macaque monkeys (*Zheng et al., 1991*; *Keller et al., 2011*). However, this was later disputed by another study, which showed no differences in vascular supply between blobs and inter-blobs in V1 (*Adams et al., 2015*).

*Figure 5a–b* shows that $R_1$ differences are in the range of around 1–2%. This is smaller but comparable to $R_1$ differences between cortical areas, which are in the range of a few percent at 7 T (*Marques et al., 2017*). The required high spatial resolution and the measured small effect sizes limited the contrast-to-noise ratio in our measurements. This most probably hindered a direct visualization of stripes at the voxel level in $R_1$ maps as illustrated in *Figure 4* (see Appendix 1 for visualization of cortical $R_1$ and $R_2^*$ in V1 and V2 for each participant) and required pooling of data within stripe types defined by fMRI followed by a statistical analysis. The coefficient of variation in $R_1$ maps was 11.3 ± 0.7 (mean ± standard deviation across participants) in V2. However, when data were pooled across participants, we could robustly detect significant differences between stripe types. We believe that the usage of qMRI parameters at this spatial scale in living humans is favorable as these parameters are less affected by technical biases and are in physical units and thus more accessible for biophysical modeling, which will facilitate studies of tissue microstructure with higher specificity.

The following considerations are related to the definition of the stripe ROIs. We could only functionally delineate thin and thick stripes but not pale stripes. Thus, the definition of pale stripes was indirect and relied on the assumption of a tripartite architecture.

In the analysis, the whole stimulated part of V2, which could be defined by retinotopy, was considered. However, the paradigms used for localization of color and disparity stripes did not show pronounced activation at the representation of the central fovea (see *Figure 2* and Appendix 1). First, the color stimulus with red/blue gratings (see *Figure 1a*) is expected to have a different effect in the central fovea than in parafoveal regions due to the macula lutea (yellow pigmented spot of the retina) and absence of blue cones in the central fovea, which might have hindered the detection of color stripes there (*Nasr and Tootell, 2018*). Second, missing activation at the representation of the central fovea for the disparity stimulus could be due to eccentricity dependence of disparity tuning. Other possible explanations might be a smaller fusion limit in central fovea or a global attention mechanism, which might preferentially activate peripheral representations (*Tsao et al., 2003*). Using conventional fMRI with lower resolution, *Tsao et al., 2003*, found an overall similar eccentricity-dependent activation pattern for a stimulation with the same maximal disparity (±0.22°). To address the possibility that our results were biased by eccentricity, we repeated the analysis shown in *Figure 5* but excluded the foveal representation of V2, which yielded basically the same results (see *Figure 5—figure supplement 1* and *Figure 5—figure supplement 3*).

*Figure 2* shows that activation maps for color-selective thin and disparity-selective thick stripes partly overlap, which might complicate the definition of separate ROIs for thin and thick stripes. It should be kept in mind that spatial overlap is expected to some degree and mainly driven by the limiting physiological point spread function of the measured blood oxygenation level-dependent signal in fMRI (*Polimeni et al., 2010*; *Chaimow et al., 2018*). This did not interfere with our analysis, since all data points with overlapping activation were excluded in ROI definitions. On the one hand, it is expected that the degree of overlap depends on the chosen $z$-score threshold level (*Nasr et al., 2016*) assuming higher thresholds to increase the probability of solely sampling within one stripe type. On the other hand, high $z$-score thresholds bear the risk to predominantly sample from large veins (*Boxerman et al., 1995*), which degrades the accuracy of the ROI due to blurring and displacement of the functional signal (*Olman et al., 2007*). We based the ROI definition on activation maps from differential contrasts between two experimental conditions as illustrated in *Figure 1a–b*, which are known to be less affected by unspecific macrovascular contributions and draining veins. Furthermore, we would have expected any venous bias to be reflected in $R_2^*$ maps (*Yacoub et al., 2001*; *Peters et al., 2007*), for example, by uneven sampling of veins between stripe types which is not the case. For these reasons, we conclude that venous bias did not drive our results.

The regular compartmentalization of V2 into distinct stripe types leads to the expectation of specific coverage of cortical area by thin, thick, and pale stripes. For example, it is expected that thick stripes are slightly larger than thin stripes as their name suggests, and that pale stripes cover around 50% of V2 (*Shipp and Zeki, 1985*; *Tootell and Hamilton, 1989*). Using fMRI for ROI definitions, the coverage depends on the chosen $z$-score threshold as stated further above. For $z = 1.96$ ($p < 0.05$, two-sided), the relative V2 coverage of non-overlapping portions of thin and thick stripes is 14.1% ± 3.4% and 24.5% ± 6.9% (mean ± standard deviation across participants and hemispheres; see Appendix 4 for absolute coverage of stripe ROIs at different threshold levels). This sums up to a pale stripes coverage of 61.4%.

An alternative hypothesis for different myelination of V2 stripes is that only the borders between pale and dark CO stripes are more strongly myelinated. *Pistorio et al., 2006*, showed in monkey V1 that the edges of blobs are more myelinated rather than blobs or inter-blob regions itself using a modified Gallyas stain. We note that the used methods in the present study neither allow a functional determination of a definite border between stripe types nor are able to spatially resolve the border region between stripes. Therefore, we cannot rule out this alternative hypothesis. However, to our knowledge, no previous study showed evidence for higher myelination of stripe borders in V2.

Measurements with high resolution are vulnerable to head movements during image acquisition, especially for the long anatomical scans. Therefore, we used an optical tracking system to prospectively correct head movements during anatomical scans (see Materials and methods). With this system, head movements could be robustly detected and corrected for at the length scale of movements induced by respiration and heart beat. Exemplary motion traces are shown in Appendix 5.

The packing density of myelinated fibers in the cerebral cortex varies with cortical depth (*Flechsig, 1920*; *Glasser and Van Essen, 2011*) and is also dependent on the cortical folding (*Smart and McSherry, 1986*). The correct and consistent sampling of data within cortex is therefore critical for

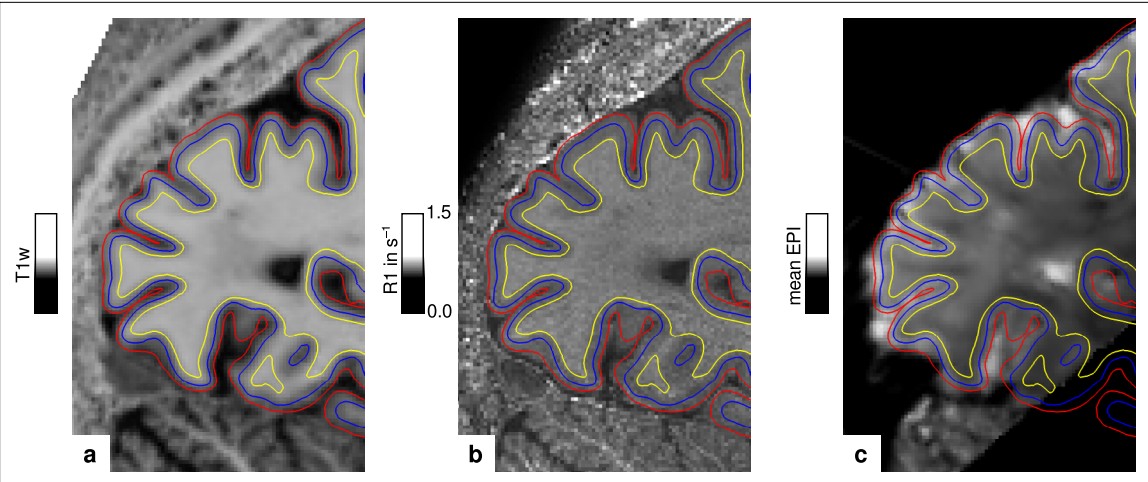

**Figure 6.** Illustration of segmentation and registration quality. (**a**) Posterior part of the MP2RAGE UNI image in sagittal orientation of one representative participant (subject 3), which was used for segmentation of the cerebral cortex. Overlaid contour lines show the reconstructed white matter/gray matter boundary surface (yellow), the pial boundary surface (red), and a surface at mid-cortical depth (blue). The computed $R_1$ map from the multi-parameter mapping (MPM) acquisition and the temporal mean from a representative functional magnetic resonance imaging (fMRI) session are shown in (**b**) and (**c**), respectively. Overlaid contour lines are identical to (**a**) to visualize the segmentation and registration quality. Other illustrations of registration quality are provided for each participant in video format, see *Figure 6—video 1*, *Figure 6—video 2*, *Figure 6—video 3*, and *Figure 6—video 4*. A more quantitative analysis of the achieved registration performance is shown in Appendix 7.

The online version of this article includes the following video(s) for figure 6:

**Figure 6—video 1.** Illustration of registration quality (subject 1).
https://elifesciences.org/articles/78756/figures#fig6video1

**Figure 6—video 2.** Illustration of registration quality (subject 2).
https://elifesciences.org/articles/78756/figures#fig6video2

**Figure 6—video 3.** Illustration of registration quality (subject 3).
https://elifesciences.org/articles/78756/figures#fig6video3

**Figure 6—video 4.** Illustration of registration quality (subject 4).
https://elifesciences.org/articles/78756/figures#fig6video4

our study. We used the equi-volume model to sample at mid-cortical depth. This model has been shown to be less affected by curvature biases than other models (e.g. equi-distant sampling) (*Waehnert et al., 2014*). The validity of the depth model also depends on accurate cortex segmentation. We visually inspected the cortical segmentation carefully in each participant (see *Figure 6*). Remaining curvature contributions were regressed out as in other studies (*Glasser and Van Essen, 2011*; *Sereno et al., 2013*; *Dumoulin et al., 2017*).

Our findings point toward higher myelination of pale stripes which exhibit lower oxidative metabolic activity according to staining with CO. V2 receives most of its input from V1 and the pulvinar (*Tootell et al., 1983*; *Sincich and Horton, 2005*). Pulvinar projections, however, only terminate in layers 3 and 5, whereas the alternating myelin pattern is most obvious in layer 4, which receives input almost exclusively from V1 (*Tootell et al., 1983*; *Sincich and Horton, 2005*). An anterograde tracer study in macaques by *Sincich and Horton, 2002*, showed that [³H]proline injections into V1 preferentially targeted V2 pale stripes. Although we cannot exclude that systematic differences in terminal axonal arborization between stripe types could explain this observation, we speculate that the results of that study correspond to higher axonal density of V1 to V2 projections in layer 4 of pale stripes. This would lead to higher myelination in pale stripes, which is in line with our measurements.

By comparing myelin-sensitive longitudinal relaxation rates ($R_1$) between stripe types in V2 defined by high-resolution fMRI, we found significant quantitative evidence for myelination differences in living humans at the level of columnar systems. This shows the feasibility to use high-resolution quantitative $R_1$ values to study cortical myelination, which is known to be less biased by technical artifacts and are thus better comparable among participants and scanner sites. Moreover, it is well known that the myelination of cortical areas and structures affects their functional properties, that is, the propagation

of action potentials (*Sanders and Whitteridge, 1946*), and correlates with postnatal development (*Glasser and Van Essen, 2011*). Therefore, the estimation of myelin content of specific structures in the human brain in vivo may increase our knowledge about its relationship to functional properties of the brain in particular and the ontogeny of the human brain in general. Our study shows that with the use of qMRI at ultra-high magnetic field strength, this is possible at the spatial scale of thin, thick, and pale stripes. We therefore believe that the current study shows the applicability of qMRI to further advance our knowledge of cortical myelination and tissue microstructure for exploration of structure-function relationships in the living human brain at mesoscopic scale.

## Materials and methods

### Participants

Four healthy participants (1 female, age = 27.50 ± 4.39, mean ± standard deviation) gave written informed consent to participate in this study. The study was approved by the local ethics committee of the University of Leipzig (reference number: 273–14). All participants had normal or corrected-to-normal visual acuity, normal color vision (based on Ishihara and Farnsworth D15 tests), and normal stereoscopic vision (based on Lang I test). For conducting the optometric tests, separate approval was granted by the ethics committee (reference number: 006–19).

### General procedures

Each participant was scanned multiple times on different days in an ultra-high field MR scanner (7 T). The first session was used to acquire a high-resolution anatomical reference scan and retinotopy data (*Sereno et al., 1995*; *Engel et al., 1997*) to functionally locate area V2 in each individual. Additionally, a baseline fMRI scan without task was acquired to aid between-session registrations (see below). Color-selective thin stripes (two sessions) and disparity-selective thick stripes (two sessions) were mapped in subsequent scanning sessions. For two participants, we had time to acquire a third thin and thick stripe session, respectively. However, we restricted the data analysis to the use of data from two sessions for consistency. Furthermore, high-resolution anatomical scans (one session) were acquired in a separate scanning session to estimate whole-brain quantitative MR relaxation parameters. A subset of acquired quantitative MR and fMRI retinotopy data was already used in other experiments (*Movahedian Attar et al., 2020*; *McColgan et al., 2021*) but was independently processed for this study.

### Visual stimulation

For the presentation of visual stimuli, we used an LCD projector (Sanyo PLC-XT20L with custom-built focusing objective, refresh rate: 60 Hz, pixel resolution: 1024 × 768), which was positioned inside the magnet room. To suppress interference with the MR scanner, the projector was placed inside a custom-built Faraday cage. Stimuli were projected onto a rear-projection screen mounted above the participants' chest inside the bore and viewed through a mirror attached to the head coil. This setup allowed the visual stimulation of around 22° × 13° visual angle. Black felt was put around the screen and all lights were turned off during experiments to mitigate scattered light reaching the participants' eyes. Experimental stimuli were written in GNU Octave (4.0.0, http://www.gnu.org/software/octave/) using the Psychophysics Toolbox (*Brainard, 1997*; *Pelli, 1997*; *Kleiner et al., 2007*) (3.0.14). A block design consisting of two experimental conditions was used for mapping color-selective thin stripes and disparity-selective thick stripes in V2, which was reported in detail previously (*Nasr et al., 2016*) and was only changed marginally for this experiment.

#### Experiment 1: Color-selective thin stripes

Stimuli consisted of isoluminant sinusoidal color-varying (red/blue) or luminance-varying (black/white) gratings as illustrated in *Figure 1a*. Gratings moved perpendicular to one of four orientations (0°, 45°, 90°, 135°) with direction reversals every 5 s and a drift velocity of 5°/s. Orientations were pseudo-randomized between blocks. A low spatial frequency (0.4 cpd) was used to mitigate linear chromatic aberration at color borders and exploit the relatively higher selectivity to color relative to luminance at this spatial scale (*Tootell and Nasr, 2017*). We point out that the appropriateness to use red and blue colors to stimulate color-selective thin stripes has already been demonstrated for macaques (*Tootell et al., 2004*; *Li et al., 2019*) and humans (*Nasr et al., 2016*). In one run, color and luminance stimuli

were both shown four times in separate blocks with a length of 30 s. Each run started and ended with 15 s of uniform gray. Ten runs were conducted in one session. During runs, participants were asked to fix their gaze on a central point (0.1° × 0.1°) and respond on a keypad when the fixation point changed its color (light green, dark green). To measure functional activation related to color, it is important to control for luminance variations across stimuli. Furthermore, isoluminance points between colors are known to change with eccentricity (*Livingstone and Hubel, 1987*; *Bilodeau and Faubert, 1997*). We used a flicker photometry (*Ives, 1907*; *Bone and Landrum, 2004*) paradigm to get isoluminance ratios between stimuli for each participant. In brief, the luminance of blue was set to 17.3 cd/m$^2$ (cf. *Li et al., 2019*). Before scanning, each participant performed a behavioral task inside the scanner in which they viewed a uniform blue flickering in temporal counter-phase with gray (30 Hz). Participants were asked to adjust the luminance of gray so that the perceived flickering was minimized using a keypad. This procedure was repeated to adjust the luminance for red and conducted at three different eccentricities (0°–1.7°, 1.7°–4.1°, 4.1°–8.3°). As expected, isoluminance ratios changed with eccentricity, which is illustrated in Appendix 6.

## Experiment 2: Disparity-selective thick stripes

Stimuli consisted of two overlaid random dot stereograms (RDSs) (*Blakemore and Julesz, 1971*) made of red and green dots on a black background (dot size: 0.1°, dot density: ~17%), respectively. Participants viewed stimuli through custom-built anaglyph spectacles using Kodak Wratten filters No. 25 (red) and 44A (cyan). In one condition, red and green RDSs performed a horizontal sinusoidal movement with temporal frequency of 0.25 Hz. Phases of red and green dots were 180° out of phase and initialized to create the perception of a 8 × 6 checkerboard moving periodically in depth (away and toward the participant), which is schematically illustrated in *Figure 1b*. Maximal disparity was set to ±0.22° (*Tsao et al., 2003*). In the other condition, static dots were presented, which were perceived as a plane at depth of the fixation point. In one run, both conditions were shown four times in separate blocks with a length of 30 s. Each run started and ended with 15 s of black background. Ten runs were conducted in one session. During runs, participants were asked to fix their gaze on a central point (0.2° × 0.2°) and respond on a keypad when the fixation point changed its form (square, circle). The luminance of red and green dots was kept low to decrease cross-talk between eyes (red dots through red filter: 3.1 cd/m$^2$, red dots through cyan filter: 0.07 cd/m$^2$, green dots through green filter: 5.7 cd/m$^2$, green dots through cyan filter: 0.09 cd/m$^2$). Luminance of green dots was doubled to approximately excite the same amount of cone photoreceptors with both colors (*Dobkins et al., 2000*).

## Retinotopic mapping

A standard phase-encoded paradigm (*Sereno et al., 1995*; *Engel et al., 1997*) was used to locate the stimulated portion of V2. Stimuli consisted of a flickering (4 Hz) black-and-white radial checkerboard restricted to a clockwise/anticlockwise rotating wedge (angle: 30°, period: 64 s) or expanding/contracting ring (period: 32 s) presented in separate runs to reveal polar angle and eccentricity maps, respectively; 8.25 cycles were shown in each run. Each run started and ended with 12 s of uniform gray background. Mean luminance was set to 44 cd/m$^2$. Participants were asked to fix their gaze on a central point during visual stimulation. No explicit task was given.

## Imaging

All experiments were performed on a 7 T whole-body MR scanner (MAGNETOM 7 T, Siemens Healthineers, Erlangen, Germany) equipped with SC72 body gradients (maximum gradient strength: 70 mT/m; maximum slew rate: 200 mT/m/s). For radio frequency (RF) signal transmission and reception, a single-channel transmit/32-channel receive head coil (Nova Medical, Wilmington, DE, USA) was used. At the beginning of each scanning session, a low-resolution transmit field map was acquired to optimize the transmit voltage over the occipital lobe.

Functional data was acquired with a 2D single-shot gradient-echo echo-planar imaging (EPI) sequence (*Feinberg et al., 2010*; *Moeller et al., 2010*). A coronal-oblique slab was imaged, which covered all stimulated portions of V2. The following parameters were used for the mapping of color-selective thin stripes, disparity-selective thick stripes, and the baseline fMRI scan without task: nominal voxel size = 0.8 mm isotropic, repetition time (TR) = 3000 ms, echo time (TE) = 24 ms, excitation flip angle (FA) = 77°, field of view (FOV) = 148 × 148 mm$^2$, 50 slices, readout bandwidth (rBW) = 1182 Hz/px, echo spacing = 1 ms,

partial Fourier = 6/8, and generalized autocalibrating partially parallel acquisition (GRAPPA) (*Griswold et al., 2002*) = 3. A slightly modified protocol was used for retinotopy measurements with the following parameter changes: voxel size = 1.0 mm isotropic, TR = 2000 ms, TE = 21 ms, FA = 68°, 40 slices, and rBW = 1164 Hz/px.

MR relaxation parameters ($R_1$, $R_2$*, $PD$) were measured with a multi-echo variable flip angle protocol for multi-parameter mapping (MPM) (*Vaculčiaková et al., 2022*). The protocol was adapted for whole-brain coverage with 0.5 mm isotropic voxel size and consisted of two multi-echo 3D fast low angle shot (FLASH) scans with $T_1$- and $PD$-weighting (T1w, PDw) plus maps of B1+ and B0. For T1w and PDw, the following parameters were used: TR = 25 ms, TE = 2.8–16.1 ms (6 equidistant echoes with bipolar readout), FA(PDw/T1w) = 5°/24°, FOV = 248 × 217 × 176 mm³ (read × phase × partition), rBW = 420 Hz/px, and GRAPPA = 2 × 2 in both phase-encoding directions. Head movements during the scan were corrected prospectively using an optical tracking system (Kineticor, USA). For motion detection, a mouth guard assembly with attached markers was manufactured for each participant by the Department of Cariology, Endodontology and Periodontology of the University of Leipzig Medical Center (*Vaculčiaková et al., 2022*). No prospective motion correction was used during functional scans because the camera system and the projection screen did not fit together in the bore. Note that functional scans are also less sensitive to motion due to the short acquisition time per volume.

For the correction of RF transmit field (B1+) inhomogeneities in relaxation parameter maps ($R_1$, $PD$), we followed the procedure detailed in *Lutti et al., 2010*; *Lutti et al., 2012*, acquiring spin-echo and stimulated echo images with a 3D EPI readout. The total scanning time of the MPM protocol was approximately 45 min.

For cortex segmentation and image registration, a whole-brain anatomy was acquired using a 3D T1-weighted MP2RAGE sequence (*Marques et al., 2010*) with the following parameters: voxel size = 0.7 mm isotropic, TR = 5000 ms, TE = 2.45 ms, inversion times (TI1/TI2) = 900 ms/2750 ms with FA = 5°/3° for T1w and PDw images, respectively, FOV = 224 × 224 × 168 mm³ (read × phase × partition), rBW = 250 Hz/px, partial Fourier = 6/8, and GRAPPA = 2 (primary phase-encoding direction; outer loop). From both inversion times, a uniform $T_1$-weighted image (UNI) and a $T_1$-map were created in the online image reconstruction on the scanner.

## Data analysis

Functional time series from color-selective and disparity-selective stripe mapping sessions were corrected for within-run and between-run motion using SPM12 (v6906, https://www.fil.ion.ucl.ac.uk/spm/) with Matlab R2019b (MathWorks, Natick, MA, USA). Motion corrected time series were high-pass filtered (cutoff frequency: 1/270 Hz) and voxel-wise statistical analyses were performed for each session using a general linear model as implemented in SPM12 with both experimental conditions as regressors.

For retinotopy measurements, time series were first converted to percent signal change by division by their temporal mean. Furthermore, slice timing correction was added before motion correction by voxel-wise temporal interpolation to a common time grid using *3drefit* from Analysis of Function NeuroImages software (*Cox, 1996*) (AFNI, 19.1.05). Motion corrected time series were high-pass filtered (cutoff frequency: 1/(3 × stimulus cycle period) Hz) and data from the first quarter of the stimulus cycle was discarded from further processing. A voxel-wise Fourier transform was computed and real and imaginary parts at stimulus frequency were averaged from runs with opposite stimulus direction to compensate for the hemodynamic lag. A phase map from averaged polar angle real and imaginary parts was computed to delineate the borders of V2.

For one participant (subject 3), we acquired a second set of retinotopy data in a separate session due to low functional responses in comparison to other participants. Motion corrected time series from the second session were registered nonlinearly to the first session using ANTs (2.3.1, http://stnava.github.io/ANTs/; *Avants et al., 2019*). Time series from both session were then averaged before further processing.

Quantitative parameter maps ($R_1$, $R_2$*, $PD$) were computed using the hMRI toolbox (*Tabelow et al., 2019*) (0.2.2, http://hmri.info) implemented in SPM12 (v7487). In brief, T1w and PDw images from the MPM protocol were averaged across echoes and used to compute a registration between both contrasts using SPM12. All available echoes from both contrasts were then used to compute an $R_2$* map by ordinary least squares regression using the ESTATICS model (*Weiskopf et al., 2014*). For

the calculation of $R_1$ and $PD$ maps, the extrapolation of T1w and PDw to TE = 0 (to remove any $R_2$*-weighting bias from resulting maps) was fit to an approximation of the Ernst equation for short-TR dual flip angle measurements using the FLASH signal (*Helms et al., 2008*; *Edwards et al., 2021*). The B1$^+$ field map was corrected for off-resonance effects using the acquired B0 map. A flip angle map was computed from the resulting B1$^+$ map to correct the apparent flip angles for inhomogeneities of the RF transmit field in the fitting procedure. For $PD$ map calculations, the resulting map was corrected for the receiver coil sensitivity profile using the adapted data-driven UNICORT method, which applies the bias field correction implemented in the segmentation module of SPM12 (*Weiskopf et al., 2011*), and calibrated such that the mean $PD$ over a white matter mask $PD$(WM) = 69 percent units (pu) (*Tofts, 2018*). Final maps ($R_1$, $R_2$*, $PD$) were corrected for spatial gradient nonlinearity distortions using the gradunwarp toolbox (*Glasser et al., 2013*; 1.0.2, https://github.com/Washington-University/gradunwarp; *Subramaniam and Brown, 2014*) and spherical harmonic coefficients provided by the manufacturer.

Cortex segmentation was based on the MP2RAGE UNI image. First, the UNI image was corrected for gradient nonlinearities with the gradunwarp toolbox and remaining bias fields using SPM12. The resulting image was input to the *recon-all* pipeline in FreeSurfer (*Dale et al., 1999*; *Fischl et al., 1999*) (6.0.0, http://surfer.nmr.mgh.harvard.edu/) with the *hires* flag to segment at the original voxel resolution (*Zaretskaya et al., 2018*). The brain mask used during segmentation was computed from the second inversion image of the MP2RAGE using SPM12 and was defined by excluding all voxels that exceeded the tissue class threshold of 10% in non-WM and non-GM tissue classes. Final gray matter/white matter and pial boundary surfaces were corrected manually. Extra care was applied to correct the pial surface around the sagittal sinus. The resulting gray matter/white matter surface was shifted 0.5 mm inward to counteract a potential segmentation bias using FreeSurfer with MP2RAGE (*Fujimoto et al., 2014*). Final surface meshes were slightly smoothed and upsampled to an average edge length of around 0.3 mm. A surface mesh at mid-cortical depth was computed using the equi-volume model (*Waehnert et al., 2014*; *Wagstyl et al., 2018*).

All images were registered to the space of the qMRI maps. For the registration of MP2RAGE and MPM, we used $R_1$ maps from both acquisitions. Just for the purpose of registration, both images were corrected for potentially remaining bias fields (SPM12) and a brain mask was applied. Images were then transformed into the same space via the scanner coordinate system and a rigid registration was computed using *flirt* (*Jenkinson et al., 2002*) (6.0) from the FMRIB Software Library (5.0.11; https://fsl.fmrib.ox.ac.uk/fsl/fslwiki/). A nonlinear transformation was computed to register activation maps and qMRI data in several steps. First, the baseline fMRI scan from the the first session was registered to the MP2RAGE using the Symmetric Normalization (SyN) algorithm (*Avants et al., 2008*) from Advanced Normalization Tools (ANTs, 2.3.1, http://stnava.github.io/ANTs/; *Avants et al., 2019*). A nonlinear registration was chosen to account for geometric distortions in functional images resulting from the low bandwidth in phase-encoding direction. Since both images were acquired in the same session, the registration between modalities was robust. Both images were prepared by removing any bias fields (*Tustison et al., 2010*) and applying a brain mask. Functional data from other sessions were then registered nonlinearly to the baseline EPI using the same procedure. The final transform was computed by concatenating transforms from all steps (EPI → baseline EPI → MP2RAGE → MPM). An exemplary illustration of the registration and segmentation quality can be seen in *Figure 6*. A more detailed analysis on achieved registration accuracy is given in Appendix 7.

Generated surfaces from cortex segmentation were transformed to MPM space using linear interpolation. For data sampling, images were transformed to MPM space using linear interpolation before sampling onto the surface mesh using nearest neighbor interpolation.

## Reliability analysis of fMRI sessions

The consistency of activation maps was analyzed by computing the vertex-wise correlation of activities within V2 between sessions acquired on different days. Spearman's rank correlation coefficient $r$ was computed. A $p$-value was determined by permutation testing. A null distribution was created by computing correlation coefficients between data from the first session and spatially shuffled data from the second session $n$ times ($n = 10,000$). We paid attention to preserve the spatial autocorrelation in spatially shuffled maps using the BrainSMASH package (*Burt et al., 2020*) (0.10.0) to consider the non-independence of data from neighboring locations. The $p$-value was then defined as the fraction

of the null distribution which is greater or smaller than $r$. We corrected the estimate of the $p$-value for the variability resulting from the finite sample size of the null distribution. The variability was described by the variance of the binomial distribution $\sigma^2 = np(1-p)$. Here, we used an upper bound of $3\sigma$, which was added to the number of samples exceeding the test statistics (*Burt et al., 2020*). A $p$-value of <0.05 was considered as statistically significant.

### Quantitative comparison of qMRI parameters between stripe types

We tested the hypothesis that pale stripes are differentially myelinated in comparison to color-selective thin and disparity-selective thick stripes. Activation maps from color and disparity stripe measurements were averaged across sessions, respectively. For participants with more than two acquired sessions, we chose to use the two sessions with highest between-session correlation of activities within V2. Color and disparity stripes were demarcated by thresholding activation maps at a selected threshold level $z \in \{0, 0.5, \ldots, 4.5\}$. Data points that did not exclusively belong to one stripe type were discarded. Similar to a procedure described in *Li et al., 2019*, mean qMRI parameter values across participants sampled in color/disparity stripes were tested against the mean throughout V2 excluding values sampled in disparity/color stripes to correct for effects from the other stripe type. This allowed us to indirectly infer effects in pale stripes assuming a tripartite stripe division of V2. For each participant, we subtracted the mean within V2 from qMRI parameter values to account for inter-subject variability (e.g. see variability between participants in Appendix 2). We considered the covariance of qMRI parameter values with local curvature of the cortical sheet (*Sereno et al., 2013*) by regressing out any linear curvature dependencies. Note that partial volume effects induced by cortical folding are themselves linear, which justifies the use of linear regression. The mean was computed across participants and statistical significance was determined by permutation testing. A null distribution was created by repeating the same procedure $n$ times ($n = 10,000$) with ROIs generated from spatially shuffled activation maps. The spatial autocorrelation in shuffled maps was preserved using the BrainSMASH package (*Burt et al., 2020*) and the $p$-value was computed as stated further above for the fMRI reliability analysis.

## Acknowledgements

The research leading to these results has received funding from the European Research Council under the European Union's Seventh Framework Program (FP7/2007-2013)/ERC grant agreement no. 616905. Nikolaus Weiskopf has received funding from the European Union's Horizon 2020 research and innovation program under the grant agreement no. 681094 and from the BMBF (01EW1711A and B) in the framework of ERA-NET NEURON. We thank the University of Minnesota Center for Magnetic Resonance Research for the provision of the multiband EPI sequence software. We thank Roland Mueller for the help with building the anaglyph spectacles.

## Additional information

### Competing interests

Nikolaus Weiskopf: The Max Planck Institute for Human Cognitive and Brain Sciences has an institutional research agreement with Siemens Healthcare. Nikolaus Weiskopf holds a patent on MRI data acquisition during spoiler gradients (United States Patent 10,401,453). Nikolaus Weiskopf was a speaker at an event organized by Siemens Healthcare and was reimbursed for the travel expenses. The other authors declare that no competing interests exist.

### Funding

| Funder | Grant reference number | Author |
|---|---|---|
| European Research Council | ERC grant agreement no 616905 | Nikolaus Weiskopf |
| Horizon 2020 - Research and Innovation Framework Programme | Grant agreement no 681094 | Nikolaus Weiskopf |

| Funder | Grant reference number | Author |
|---|---|---|
| Bundesministerium für Bildung und Forschung | 01EW1711A & B | Nikolaus Weiskopf |

The funders had no role in study design, data collection and interpretation, or the decision to submit the work for publication. Open access funding provided by Max Planck Society.

## Author contributions
Daniel Haenelt, Conceptualization, Data curation, Software, Formal analysis, Investigation, Visualization, Methodology, Writing – original draft, Writing – review and editing; Robert Trampel, Supervision, Investigation, Writing – review and editing; Shahin Nasr, Jonathan R Polimeni, Roger BH Tootell, Martin I Sereno, Software, Methodology, Writing – review and editing; Kerrin J Pine, Investigation, Methodology, Writing – review and editing; Luke J Edwards, Methodology, Writing – review and editing; Saskia Helbling, Formal analysis, Writing – review and editing; Nikolaus Weiskopf, Conceptualization, Resources, Supervision, Funding acquisition, Project administration, Writing – review and editing

## Author ORCIDs
Daniel Haenelt http://orcid.org/0000-0003-2310-5086
Robert Trampel http://orcid.org/0000-0002-3080-9412
Jonathan R Polimeni http://orcid.org/0000-0002-1348-1179
Martin I Sereno http://orcid.org/0000-0002-7598-7829
Luke J Edwards http://orcid.org/0000-0002-8320-7298
Nikolaus Weiskopf http://orcid.org/0000-0001-5239-1881

## Ethics
Human subjects: Participants gave written informed consent to participate in this study. The study was approved by the local ethics committee of the University of Leipzig (reference number: 273-14). For conducting the optometric tests, separate approval was granted by the ethics committee (reference number: 006-19).

## Decision letter and Author response
Decision letter https://doi.org/10.7554/eLife.78756.sa1
Author response https://doi.org/10.7554/eLife.78756.sa2

# Additional files

## Supplementary files
• MDAR checklist

## Data availability
Anonymized and defaced MRI data used in the present study are openly accessible at: https://osf.io/624cz/.

The following dataset was generated:

| Author(s) | Year | Dataset title | Dataset URL | Database and Identifier |
|---|---|---|---|---|
| Haenelt D, Trampel R, Nasr S, Polimeni JR, Tootell RBH, Sereno MI, Pine KJ, Edwards LJ, Helbling S, Weiskopf N | 2022 | High resolution quantitative and functional MRI indicate lower myelination of thin and thick stripes in human secondary visual cortex | https://doi.org/10.17605/OSF.IO/624CZ | Open Science Framework, 10.17605/OSF.IO/624CZ |

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

## Appendix 1

**Appendix 1—figure 1.** Individual functional magnetic resonance imaging (fMRI) activation maps and quantitative magnetic resonance imaging (qMRI) maps (subject 1). The first row shows activation maps for color-selective thin stripe measurements (contrast: color > luminance) (**a, c**) and disparity-thick-stripe measurements (contrast: depth > no depth) (**b, d**). The second row shows eccentricity (**e, g**) and polar angle (**f, h**) phase responses from the retinotopy measurement, which were used to define the positions of V1/V2 and V2/V3 borders. The third row shows qMRI maps of $R_1$ (**i, l**) and $R_2$* (**k, m**) from the multi-parameter mapping (MPM) acquisition, which were used to compare myelin content between stripe types. Other details as in *Figure 2*. Scale bar: 1 cm.

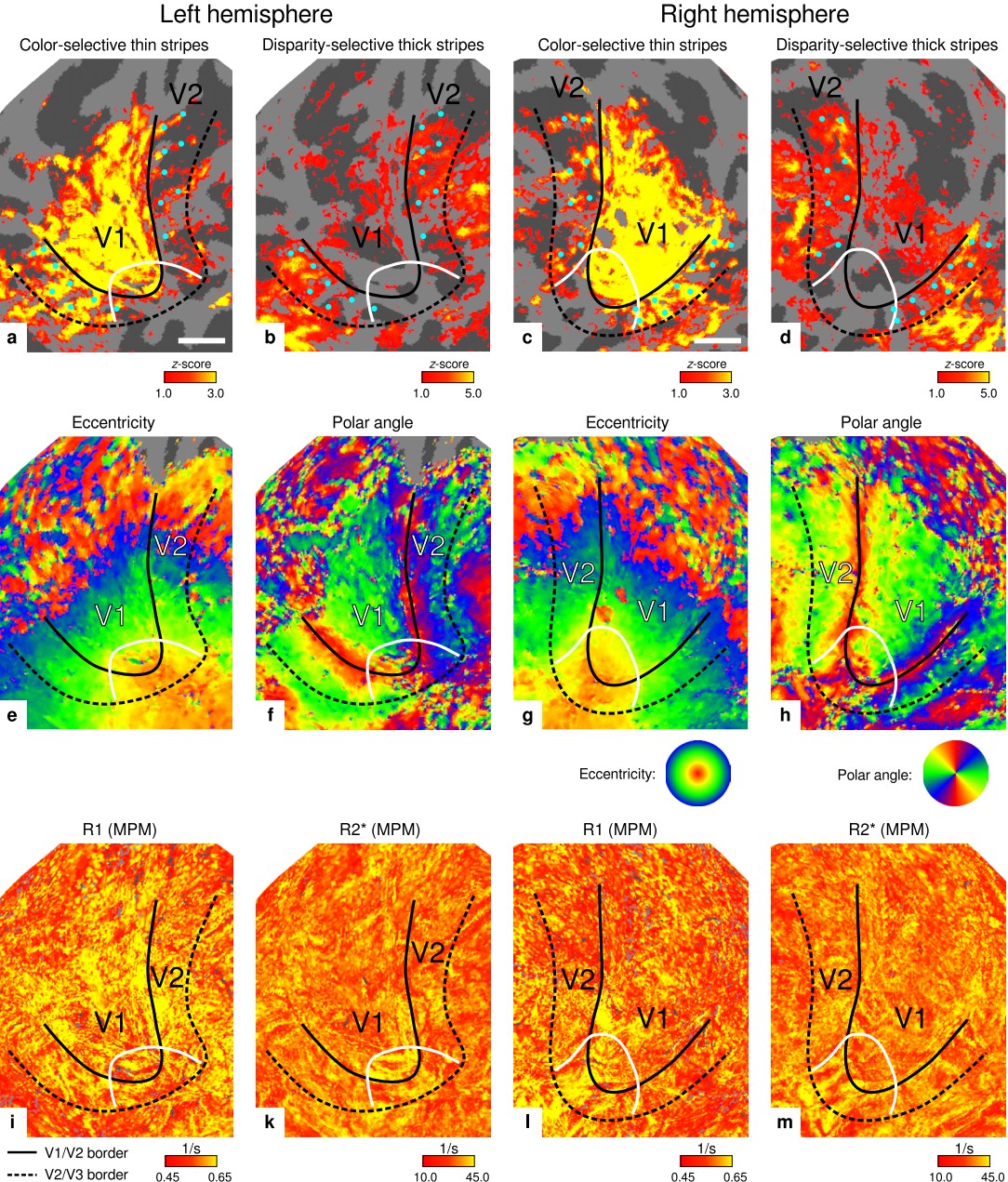

**Appendix 1—figure 2.** Individual functional magnetic resonance imaging (fMRI) activation maps and quantitative magnetic resonance imaging (qMRI) maps (subject 2). The first row shows activation maps for color-selective thin stripe measurements (contrast: color > luminance) (**a, c**) and disparity-thick-stripe measurements (contrast: depth > no depth) (**b, d**). The second row shows eccentricity (**e, g**) and polar angle (**f, h**) phase responses from the retinotopy measurement, which were used to define the positions of V1/V2 and V2/V3 borders. The third row shows qMRI maps of $R_1$ (**i, l**) and $R_2^*$ (**k, m**) from the multi-parameter mapping (MPM) acquisition, which were used to compare myelin content between stripe types. Other details as in *Figure 2*. Scale bar: 1 cm.

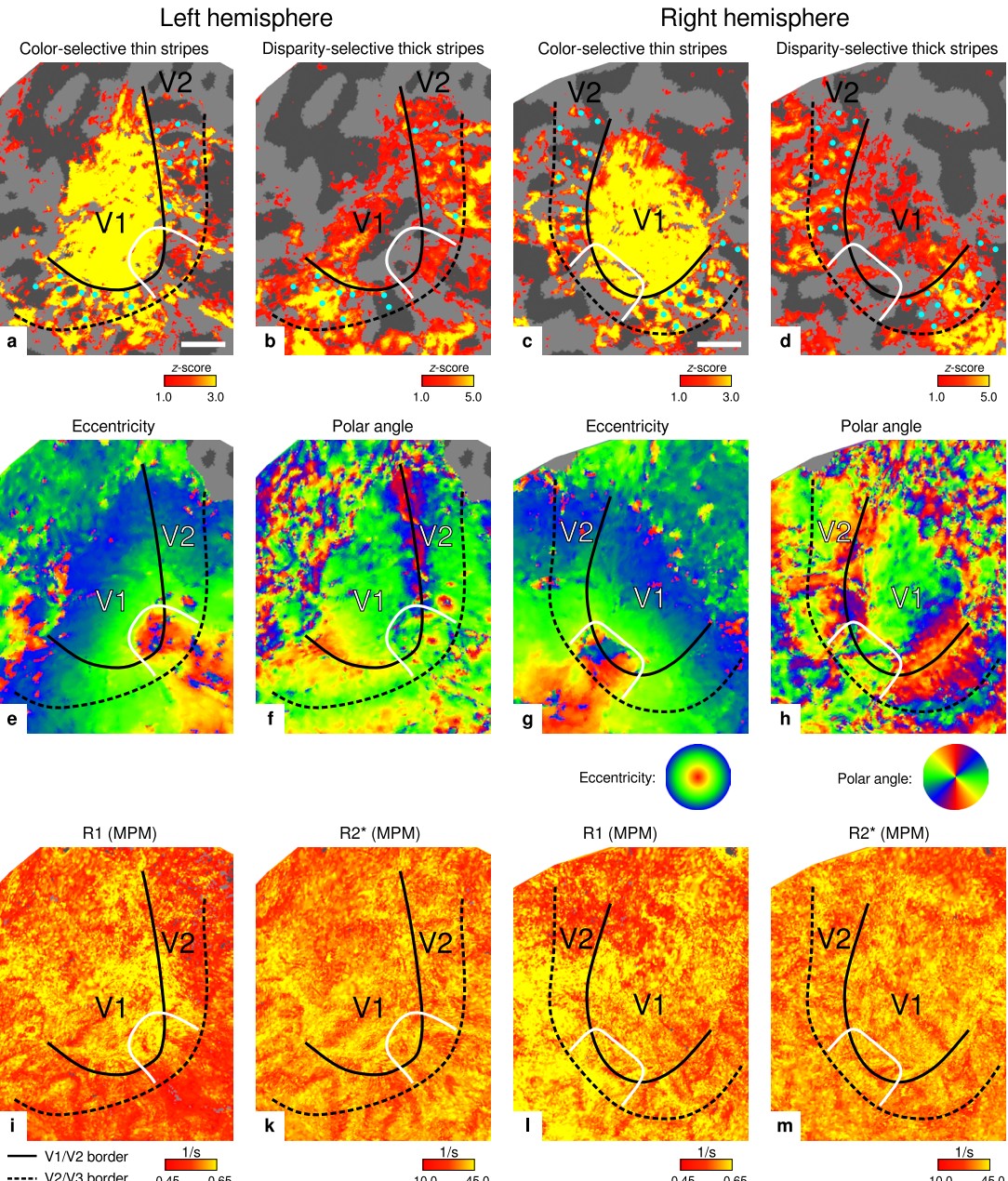

**Appendix 1—figure 3.** Individual functional magnetic resonance imaging (fMRI) activation maps and quantitative magnetic resonance imaging (qMRI) maps (subject 3). The first row shows activation maps for color-selective thin stripe measurements (contrast: color > luminance) (**a, c**) and disparity-thick-stripe measurements (contrast: depth > no depth) (**b, d**). The second row shows eccentricity (**e, g**) and polar angle (**f, h**) phase responses from the retinotopy measurement, which were used to define the positions of V1/V2 and V2/V3 borders. The third row shows qMRI maps of $R_1$ (**i, l**) and $R_2^*$ (**k, m**) from the multi-parameter mapping (MPM) acquisition, which were used to compare myelin content between stripe types. Other details as in *Figure 2*. Scale bar: 1 cm.

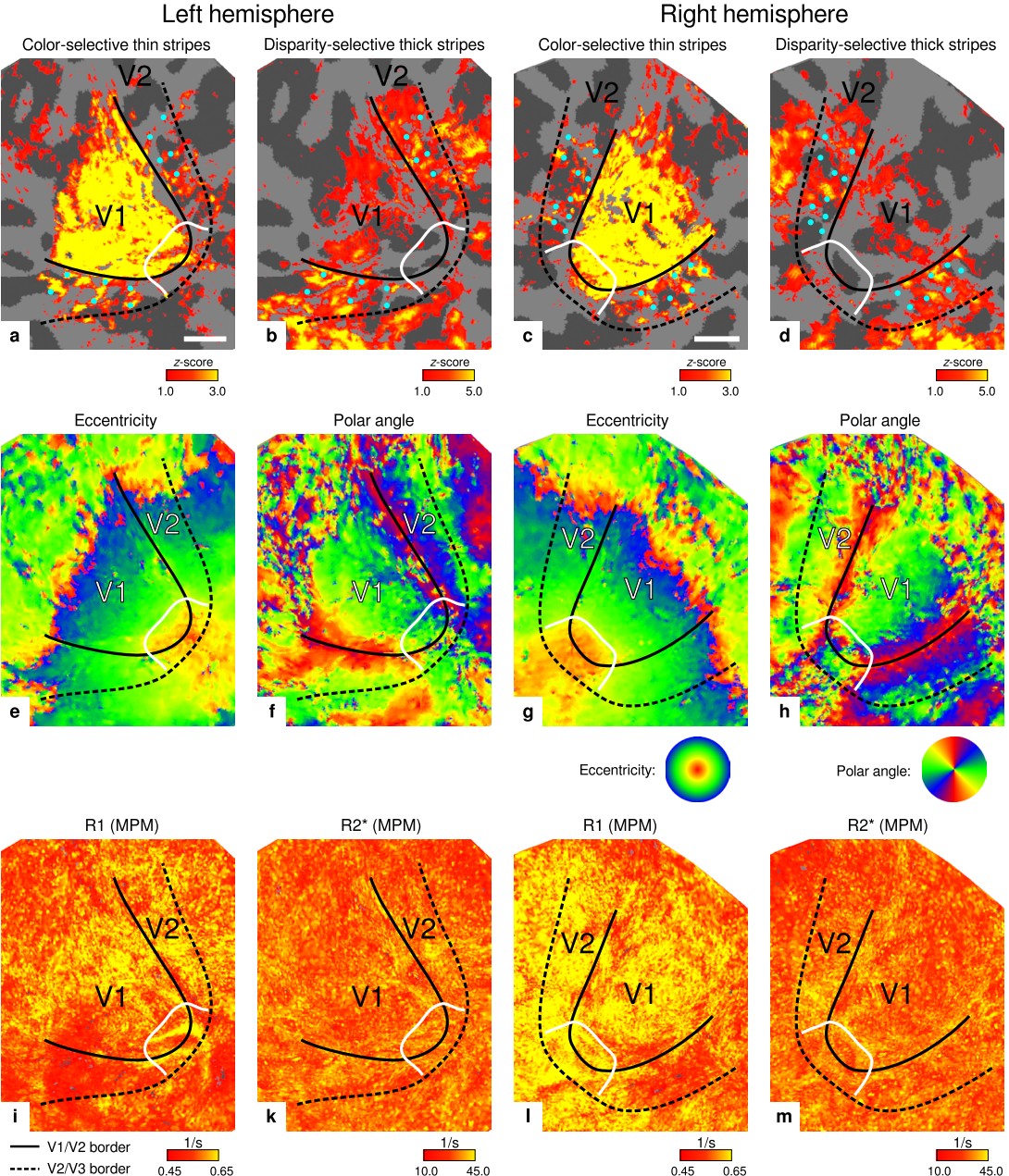

**Appendix 1—figure 4.** Individual functional magnetic resonance imaging (fMRI) activation maps and quantitative magnetic resonance imaging (qMRI) maps (subject 4). The first row shows activation maps for color-selective thin stripe measurements (contrast: color > luminance) (**a, c**) and disparity-thick-stripe measurements (contrast: depth > no depth) (**b, d**). The second row shows eccentricity (**e, g**) and polar angle (**f, h**) phase responses from the retinotopy measurement, which were used to define the positions of V1/V2 and V2/V3 borders. The third row shows qMRI maps of $R_1$ (**i, l**) and $R_2$* (**k, m**) from the multi-parameter mapping (MPM) acquisition, which were used to compare myelin content between stripe types. Other details as in *Figure 2*. Scale bar: 1 cm.

# Appendix 2

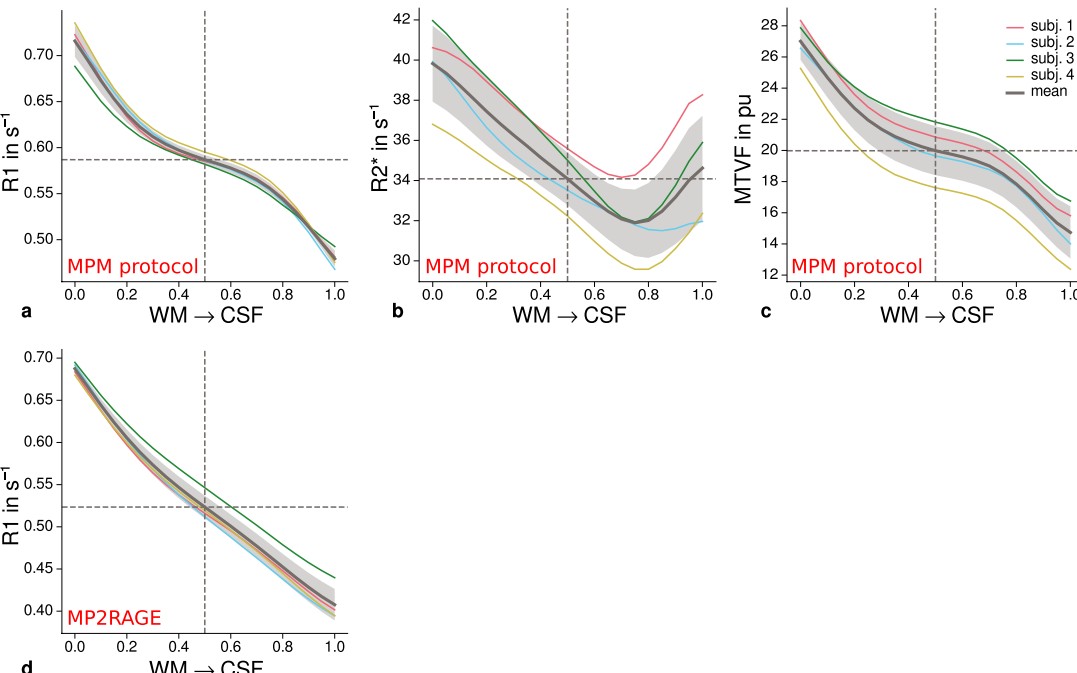

**Appendix 2—figure 1.** Cortical profiles of mean quantitative magnetic resonance imaging (qMRI) parameters within V2. Mean $R_1$ (**a**), $R_2^*$ (**b**), and *MTVF* = 100% − *PD* (macromolecular tissue volume fraction [***Mezer et al., 2013***]) (**c**) values based on the multi-parameter mapping (MPM) protocol were sampled at 21 cortical depths between the gray matter/white matter boundary surface and pial surface. The expected decrease of quantitative parameters toward the pial surface can be seen in all plots. (**d**) shows mean $R_1$ across cortical depth based on a separate data set using the MP2RAGE sequence. A similar decrease as shown in (**a**) can be seen. However, in comparison to (**a**), a plateau at middle depths is less visible. This might be attributable to the larger voxel size in (**d**) and the larger point spread function along the phase-encoding direction (***Marques et al., 2010***) of MP2RAGE acquisitions. The mean across participants is shown as gray line. Dashed lines mark the mean at mid-cortical depth. Shaded areas indicate 1 standard deviation across participants.

# Appendix 3

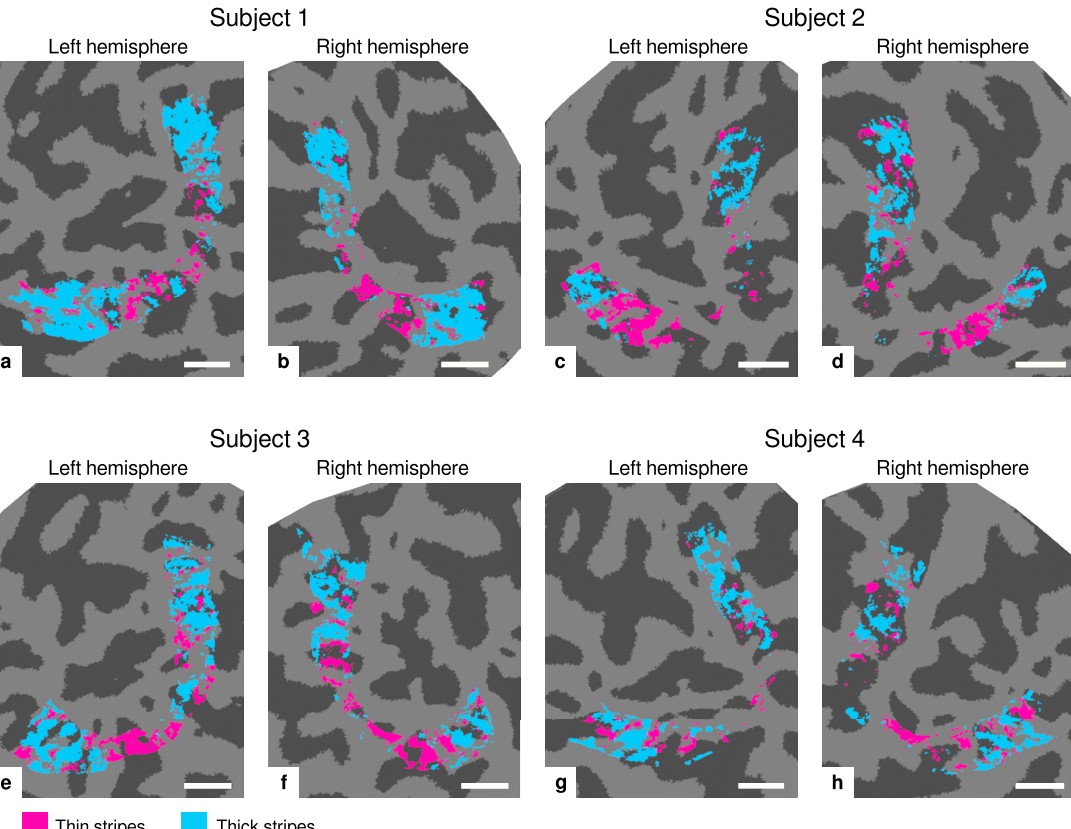

**Appendix 3—figure 1.** Visualization of thin and thick stripe regions of interest (ROIs). Thin stripe (magenta) and thick stripe (cyan) ROIs are shown for each participant. Presented ROIs were created by thresholding functional activation maps from color-selective thin stripe and disparity-selective thick stripe measurements at a $z$-score threshold level of $z = 1.96$ (p < 0.05, two-sided). Note that ROIs were defined to be mutually exclusive, that is, data points (vertices) were only classified as belonging either to thin stripes or thick stripes, if the corresponding contrast exceeded the defined threshold but not the other. To aid comparison, all maps are presented in the same format as in *Figure 2* and Appendix 1. Cortical curvature is shown in gray (sulcal cortex is dark gray and gyral cortex light gray). Scale bar: 1 cm.

## Appendix 4

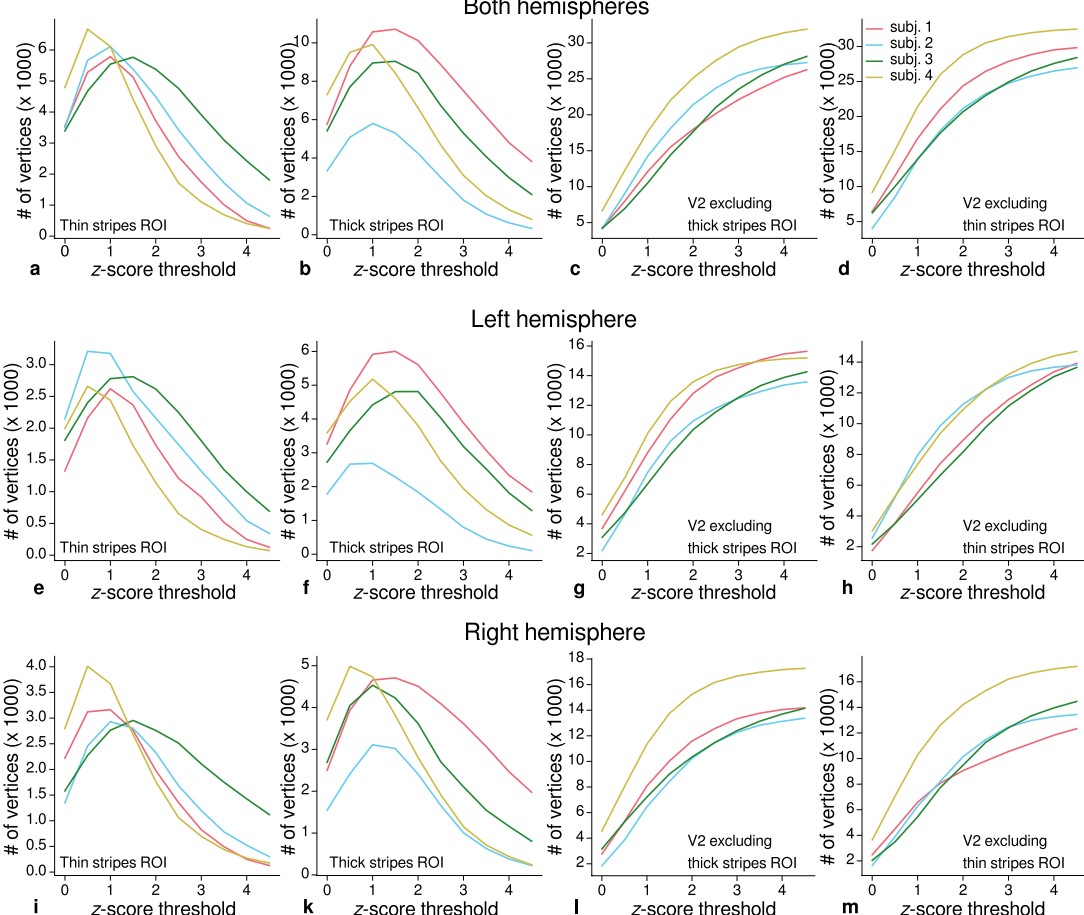

**Appendix 4—figure 1.** Sizes of stripe regions of interest (ROIs) for each threshold level. ROI sizes (number of vertices) across hemispheres for color-selective thin stripes (**a**), disparity-selective thick stripes (**b**), and whole V2 excluding one stripe type (**c–d**) are shown for several $z$-score threshold levels and all participants (upper row). These ROIs were used for comparison of quantitative magnetic resonance imaging (qMRI) parameters ($R_1$, $R_2$*) between stripe types as shown in *Figure 5*. Voxels activated by both color and disparity contrasts were excluded from thin and thick stripe ROIs, explaining the non-monotonous dependence on the $z$-score threshold. Middle and lower rows show number of vertices from single hemispheres.

## Appendix 5

### Motion trace segments from one T1w scan

**Appendix 5—figure 1.** Exemplary motion data from one multi-parameter mapping (MPM) measurement. For one representative participant (subject 3), head movements are shown from the 3D multi-echo fast low angle shot (FLASH) scan with $T_1$-weighting (T1w). Movements were measured during acquisition using an optical tracking system and used for prospective motion correction. From the start (**a**), the middle (time of $k$-space center sampling) (**b**) and the end of the scan (**c**), excerpts of 20 s depict translational movements for each TR = 25 ms in $z$-direction (inferior-superior) relative to start of the scan. In all plots, the participant's breathing pattern can be seen as slow oscillation. On top of this oscillation, displacements at a faster rate related to the cardiac cycle can be identified. In (**a**), a subset of cardiac beats are marked by red arrows. This demonstrates the ability to detect small movements, which were prospectively corrected during high-resolution anatomical scans. Note that motion was overall on the order of the (0.5 mm)³ voxel size (even for the short excerpts shown), which makes an accurate correction for subject motion necessary during the long scan time (acquisition time around 18 min). This is qualitatively illustrated by showing the 0.5 mm voxel edge length as blue square in (**b**). It can be further noticed that movements were getting larger toward the end of the scan as expected due to the long scan time.

## Appendix 6

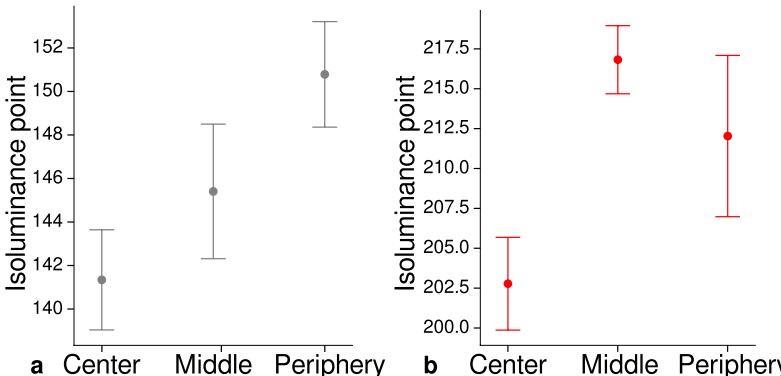

**Appendix 6—figure 1.** Measured mean isoluminance points across participants and sessions at different eccentricities. For each participant, isoluminance points were measured before scanning within the scanner at three different eccentricities (center: 0°–1.7°, middle: 1.7°–4.1°, periphery: 4.1°–8.3°). Luminance points for gray (**a**) and red (**b**) were adjusted to match blue (RGB: 0, 0, 255), see Materials and methods. Across participants and scanning sessions, we found a statistically significant effect of eccentricity for gray ($F(2, 93) = 3.25$, $p = 0.04$, $\eta_p^2 = 0.07$) and red ($F(2, 93) = 3.96$, $p = 0.02$, $\eta_p^2 = 0.08$) with the application of a one-way ANOVA. Error bars indicate 1 standard error of the mean.

## Appendix 7

### Analysis of registration accuracy

Accurate registration between qMRI and fMRI data is of utmost importance for the current study. However, knowledge about the quality of registration beyond visual inspection, which is subjective, qualitative, and not usable for large quantities of data, is difficult to achieve (*Pluim et al., 2016*).

In the following analysis, we quantitatively assessed the quality of registration by exploiting that both qMRI and fMRI data were registered to the same whole-brain anatomy (MP2RAGE). Knowing that in all acquired MR images, gray matter (GM) and white matter (WM) can easily be differentiated, we estimated registration accuracy by the analysis of local image contrast of the registered image around the GM/WM border, which was defined by cortex segmentation based on the MP2RAGE. We expect that the local image contrast is higher at the WM/GM border than in nearby locations in the registered image if registration between whole-brain anatomy and qMRI and fMRI is accurate, respectively. We restricted the analysis to vertices $v$ in V2. For each vertex $v$, we computed a normal vector and defined 100 points along the normal in equidistant steps around the GM/WM border ($\pm 10$ mm). Let $\lambda$ be the parameter defining the position on the normal line, then $v(\lambda)$ is the coordinate of vertex $v$ shifted along the normal direction by $\lambda$. For each $v(\lambda)$, we sampled data from two points with  distance along the normal vector in both positive $v_+$ and negative $v_-$ direction using trilinear interpolation. To evaluate the local image contrast, we used the cost function $J$, which was introduced for the boundary-based registration (BBR) method (*Greve and Fischl, 2009*). $J$ has the property to approach zero if local image contrast increases. Note that we did not use the BBR method for actual registration. To derive $J$, a contrast measure $Q(\lambda)$ is first computed for each $v(\lambda)$ in V2.

$$Q(\lambda) = 100 \frac{v_+(\lambda) - v_-(\lambda)}{0.5 \cdot (v_+(\lambda) + v_-(\lambda))} \tag{1}$$

The sign of the contrast depends on the imaging modality, that is, if WM is darker or brighter than GM. This was considered by choosing the direction of the normal vector to point toward the brighter tissue component. $J$ was then computed based on all contrast measures in V2.

$$J(\lambda) = \frac{1}{N} \sum_{v(\lambda) \in V2} \left[ 1 + \tanh(0.5 \cdot Q(\lambda)) \right] \tag{2}$$

where $N$ is the number of vertices in V2. *Appendix 7—figure 1* shows for each participant and imaging modality the size of $J$ around the GM/WM border. It can be seen that the position of minimum $J$ generally coincides with the GM/WM border for each data set. This enables us to conclude that accurate registration could be achieved for both qMRI and fMRI data, respectively. However, a small shift was also observed in all data sets relative to the GM/WM border. While partial volume effects could have contributed to this bias, we note that we applied an inward shift (0.5 mm) to the GM/WM boundary surface during segmentation (see Materials and methods). This was done to counteract a potential bias in the segmentation pipeline using the flat image from the MP2RAGE acquisition which yielded visually better results. The current analysis reveals that this step was probably not necessary. However, note that this did not affect any conclusions drawn in further analyses, since the shift is negligible small when compared to changes in local image contrast as shown in *Appendix 7—figure 1*.

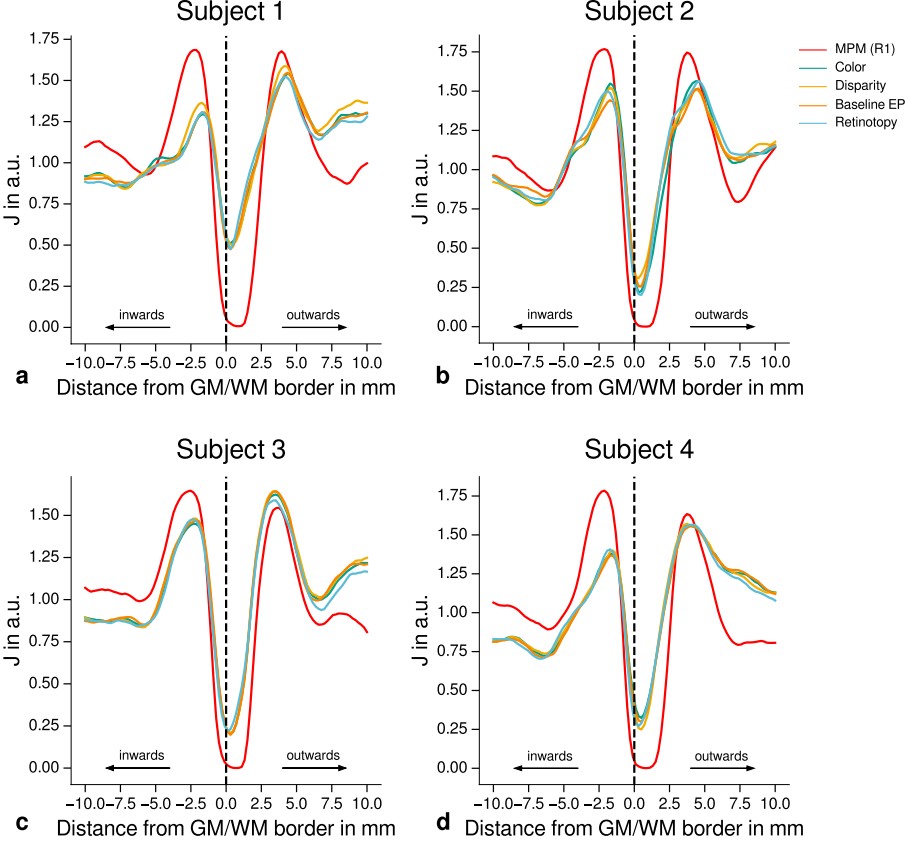

**Appendix 7—figure 1.** Quantitative analysis of registration accuracy. For each participant, the local image contrast is estimated by the cost function $J$ at multiple positions around the gray matter (GM)/white matter (WM) border. For all imaging modalities, contrast is maximal ($J$ is minimal) at the GM/WM border. This demonstrates the high quality of registration between different modalities (quantitative magnetic resonance imaging [qMRI], functional magnetic resonance imaging [fMRI]) for all participants and used data sets. The position of the GM/WM border is indicated as vertical dashed line.

