## [Editor Report]

Using a combination of cutting-edge high-resolution magnetic resonance protocols, this important study investigates the structure-function relationship of specialised compartments in the human visual cortex in vivo. The use of quantitative MRI provides convincing evidence that color- and disparity-selective functional compartments (thin and thick stripes) of visual area V2 have different MR relaxation properties than the pale stripes of V2. While these results indicate different patterns of myelination across the "stripes" of V2, the MR signals will require independent validation to be considered specific to myelin. This study will be of interest to a wide range of neuroscientists and clinicians employing imaging methods to study cortical organization.

---

## [Decision Letter]

**Decision letter after peer review:**

Thank you for submitting your article "High resolution quantitative and functional MRI indicate lower myelination of thin and thick stripes in human secondary visual cortex" for consideration by *eLife*. Your article has been reviewed by 3 peer reviewers, including Kristine Krug as the Reviewing Editor and Reviewer #1, and the evaluation has been overseen by Floris de Lange as the Senior Editor. The following individual involved in review of your submission has agreed to reveal their identity: Wim Van Duffel (Reviewer #3).

Essential revisions:

1) Specificity of the used MR signals to myelin:

– Please temper claims to the specificity of the MR protocols used throughout the manuscript and discuss the evidence critically.

– Where possible, add a quantitative comparison of different MR signals for myelin. We were also wondering whether the authors had Tw1/Tw2 maps available from the same brains.

– Discuss the literature with regard to quantitative MRI, myelin-related MR signals and histology in a more differentiated manner.

2) Visualise ROIs used for the functional stripe compartments and relate them directly to flat maps of R1 and z-scores:

– What do ROIs created from myelin-related MR measures look like? Can they be used to define stripes?

– The reviewers would like to see the three-way ROIs for all hemispheres included.

– Provide additional controls that any irregular distribution of functional ROIs throughout the visual field map has not affected the results.

– Can you exclude the possibility that the high myelination might be located at stripe borders?

3) Include supplements for all subjects showing (i) goodness of registration and (ii) how the borders of V2 were defined based on retinotopy.

*Reviewer #1 (Recommendations for the authors):*

1) One critical question is whether the direct association of the pale stripe region with higher density of myelination is safe. Ideally, the pale stripes would have been defined functionally as well and the interdigitating pattern of the three stripe compartments would have been similarly distributed throughout the imaged parts of V2. The authors should at least add the following information/analysis, please:

– Show the detailed ROI patterns (thick, thin _and_ inter stripe) used for the analysis of myelin in individual hemispheres for all subjects. This should show the regular distribution of inter stripe regions through the imaged and analysed visual field representations.

– Show that the exclusion of foveal regions from the analysis (mostly without significant activation for binocular disparity (thick stripes, all subjects) or for colour (thin stripes, subject 2)) does not affect the main conclusions. Explanation in lines 233-236 is not sufficient to exclude this.

– Discuss whether you can exclude the alternative interpretation that myelination is particularly prominent as the borders of certain stripes, as it as been indicated for V1 between patches and inter-patches (Pistorio et al. 2006). This might also explain contradictory results between papers.

– A brief discussion should be included on the adequacy of all the chosen visual stimuli, specifically for

(i) activating (or not) foveal representations of V2 (add spatial scale to the discussion of absence of blue cones in the fovea), especially for binocular disparity.

(ii) the artefactual luminance responses that red-blue gratings can generate even at isoluminance (because of chromatic aberrations: red and blue light cannot be in focus on the retina at the same time) and whether this could have activated for instance pale stripes,

and hence (iii) whether one could reasonably expect that the chosen stimuli will not activate pale stripes or whether there might be overlap (with specific reference also to the chosen binocular disparity stimulus).

2) Reconsider the easy, unnecessary straw-man of "all histological results are conflicting" in abstract and introduction for a more considered analysis of the discussed papers and of what is the most likely "ground truth" that you would expect to find with MRI. For instance, Luxol not only stains phospholipids of the neuronal myelin sheath but is also associated with staining mitochondria (Takaya, 1967), which might lead to differential results in cortical staining for CO-rich compartments. Furthermore, the contradictory results for myelin staining "of" CO compartments in the literature might also have to do with more recent carefully analysis showing that myelin might be particularly associated with borders of CO blobs and inter blobs due to their relationship with OD columns (Pistorio et al. 2006). On close inspection, some of the earlier papers (e.g. Tootle et al. 1983) present little clear data on the alignment of the different maps in the papers; even with the more recent, careful studies, one needs to take a close look at the conditions tested (e.g. monocular enucleation, Horton & Hocking 1997). The discussion (e.g. lines 183 -185) seems to be in parts more considered.

*Reviewer #2 (Recommendations for the authors):*

The Introduction strongly gives the impression that quantitative MRI emerged in the mid-2010s, and entirely from the co-authors laboratory.

On p10, the statement "To detect signal changes with magnitudes of a few percent, qMRI is advantageous since it is less biased by inhomogeneities in the radiofrequency transmit and receive fields, and therefore allows better comparison of differences between ROIs and across participants" doesn't make sense to me. I gather the authors intend this to be taken as an argument for qMRI rather than non-quantitative "weighted" imaging. There are a few oddities of logic here. First, the fact that these differences are a few percent in units of R1 does not necessarily mean they are a few percent in terms of weighted contrast; second, even if the signals are of similar magnitude of percentage, the process of quantification "introduces" noise (e.g., involves propagation of error that amplifies noise from the weighted images) making it harder to detect those small percent signals; third, this statement conflates effect size (% signal) with sources of bias. I can believe that avoiding sources of bias is a major argument for qMRI, but this statement needs editing.

On p10, the authors note that the small effect sizes "hindered a direct visualization of stripes at the voxel level in R1 maps" and refer to Figure 4. But this cannot be seen in Figure 4. I recommend adding a direct comparison of flat maps of R1 and z-score so readers can see what the data looks like. Ideally this would superimpose some of the structure from fMRI (potentially the blue dots from Figure 2, although these may obscure any structure that is present).

---

## [Author Response]

Essential revisions:(1) Specificity of the used MR signals to myelin:– Please temper claims to the specificity of the MR protocols used throughout the manuscript and discuss the evidence critically.

Thank you for this comment. We agree that we have inadvertently overstated the specificity of R1 at several occasions in the text and modified the statements in the revised manuscript accordingly (p. 2, lines 68–88; p. 6, line 163; p. 8, line 216; p. 9, lines. 257–260). Please note that we also discuss this point in the author response to reviewer #2 (public review) in more detail.

– Where possible, add a quantitative comparison of different MR signals for myelin. We were also wondering whether the authors had Tw1/Tw2 maps available from the same brains.

In the manuscript, we performed the analysis with two independent approaches (MPM and MP2RAGE) which yielded basically the same results (see Figure 5 and Figure 5-Supplement 2). Both MPM and MP2RAGE are well suited for measurements at 7 T. Unfortunately, we do not have T1w/T2w maps available for any of our participants. We also would like to point out that the acquisition of high-resolution T2-weighted images or T2 maps are very challenging at high magnetic fields and to the best of our knowledge, at 7 T, neither T2w nor T2 maps have been successfully acquired with an isotropic resolution of (0.5 mm)^3^ as used in the present study. Therefore, we were unfortunately not able to acquire T1w/T2w data at 7 T with a comparable resolution. Please also see the author response to reviewer #3 (recommendations for the authors; comment 1), which provides an extended discussion, including a discussion of different MR contrasts for myelin mapping.

– Discuss the literature with regard to quantitative MRI, myelin-related MR signals and histology in a more differentiated manner.

In the revised manuscript, we removed the phrasing that gave the impression that MRI can resolve the conflicting results of histological studies. In the introduction, we changed the corresponding paragraph by emphasizing the alternative view, which can be obtained from MRI by the possibility to investigate structure-function relationships in the living human brain (see p. 2, lines 68–70; p. 3, lines 93–95). We also included references in the introduction which show the increasing body of literature using R1 as surrogate for cortical myelination while still pointing out the limitations concerning the specificity of R1 (p. 2, lines 78–85). We also included an extended discussion about current knowledge of V2 stripes myelination in the Discussion section (pp. 8–9, lines 216–240).

(2) Visualise ROIs used for the functional stripe compartments and relate them directly to flat maps of R1 and z-scores:

We added Appendix 1, which shows figures of thresholded functional activation maps (color-selective thin stripes and disparity-selective thick stripes), retinotopy data (polar angle and eccentricity maps) and qMRI parameter maps (R1 and R2*) for each participant. Within a single hemisphere of each participant, all figures are aligned to each other, which allows direct visual comparison between maps.

– What do ROIs created from myelin-related MR measures look like? Can they be used to define stripes?

Appendix 1 of the revised manuscript shows quantitative parameter maps (R1 and R2*) for each participant in V2. Those maps do not allow direct identification of thin, thick and pale stripes. This is in line with the rather small (but reliable) R1 differences found in the current study, which demanded a statistical analysis across participants.

– The reviewers would like to see the three-way ROIs for all hemispheres included.

Since ROIs were based on functional activation maps, actual ROI sizes depend on the chosen z-score threshold level. This was the reason for performing the main analysis at different threshold levels. For visualization purposes we now illustrate the ROIs for a specific intermediate threshold level (z=1.96) in Appendix 3.

– Provide additional controls that any irregular distribution of functional ROIs throughout the visual field map has not affected the results.

As pointed out by the reviewers and editors, a consistent pattern in functional activation maps across participants was the general absence of activation around the foveal representation. Possible explanations were added to the Discussion section (pp. 9–10, lines 289–303). See also the extended discussion in the authors responses to reviewer #1 (recommendations for the authors). Since we used activation maps for ROI definitions, this could potentially have introduced a visual field dependent bias in our analysis. To exclude this possibility, we included a supporting analysis, in which we excluded the region around the foveal representation from the analysis – see further details in the revised manuscript (p. 8, lines 189–202). Excluded regions were delineated in Figure 2, Figure 2-Supplement 1 and Appendix 1 using white lines. In Figure 5-Supplement 1 and Figure 5-Supplement 3, results from this supporting analysis are shown which reproduced the primary findings from the main analysis, particularly the relatively higher myelination of pale stripes, and thus corroborate the main findings of the current study.

– Can you exclude the possibility that the high myelination might be located at stripe borders?

Thank you for this comment. This is an intriguing alternative hypothesis. Due to the limited spatial resolution of our methods, we cannot exclude this possibility. We added a paragraph in the Discussion section (p. 10, lines 330–337) to inform the reader about this alternative hypothesis. However, since the recommended article by the reviewers (Pistorio et al., 2006) only considers the CO-rich blob structure in V1 and no evidence for this hypothesis is known to the authors for area V2, we consider this hypothesis as rather speculative.

(3) Include supplements for all subjects showing (i) goodness of registration and (ii) how the borders of V2 were defined based on retinotopy.

(i) To further demonstrate the high quality of registration in the current study, we included video files showing the registration results for each participant (see Figure 6-video 1, Figure 6-video 2, Figure 6-video 3, Figure 6-video 4). We additionally included an analysis in Appendix 7, which examined the achieved goodness-of-registration.

(ii) In Figure 2—figure supplement 1, we show retinotopy data to illustrate the definition of retinotopic borders in Figure 2. In Appendix 1, retinotopy data is shown (next to the illustration of qMRI parameter maps and functional activation maps) for each participant.

Reviewer #1 (Recommendations for the authors):(1) One critical question is whether the direct association of the pale stripe region with higher density of myelination is safe. Ideally, the pale stripes would have been defined functionally as well and the interdigitating pattern of the three stripe compartments would have been similarly distributed throughout the imaged parts of V2. The authors should at least add the following information/analysis, please:– Show the detailed ROI patterns (thick, thin _and_ inter stripe) used for the analysis of myelin in individual hemispheres for all subjects. This should show the regular distribution of inter stripe regions through the imaged and analysed visual field representations.

We agree that illustration of the actual ROIs will be beneficial for the reader. We included Appendix 3, which presents ROIs at an intermediate threshold level of z=1.96 where clear differences between thick/thin and pale stripes were found (Figure 5). Please note that ROIs were defined at different threshold levels in the analysis to avoid bias due to an arbitrary choice of threshold and demonstrate robustness of the results. In the present study, activation for disparity contrasts was overall higher than for color contrasts (see e.g. that disparity contrast was shown with different color bars in Figure 2 and Appendix 1). This is also reflected in the functionally delineated ROIs, which were based on using the same threshold level for both color and disparity contrast resulting in larger areas covered by the disparity-selective ROIs in V2. Despite this, color-selective ROIs were visible throughout the entire stimulated region of V2.

– Show that the exclusion of foveal regions from the analysis (mostly without significant activation for binocular disparity (thick stripes, all subjects) or for colour (thin stripes, subject 2)) does not affect the main conclusions. Explanation in lines 233-236 is not sufficient to exclude this.

Thank you for raising this very important point. To identify and exclude this potential bias in our results, we performed a supporting analysis where we excluded the region encompassing the foveal representation in V2 (see p. 8, lines 189–202; p. 10, lines 301–303). To have a consistent exclusion criterion across participants, we used the eccentricity maps from retinotopy measurements for the definition of exclusion masks. The excluded area is illustrated by a white border in Figure 2, Figure 2-Supplement 1 and Appendix 1. Repeating the comparison of R1 and R2* between stripe types after exclusion of the foveal region corroborated the main results, i.e., significant R1 differences between thick/thin and pale stripes. This can be seen in Figure 5-Supplement 1 and Figure 5-Supplement 3.

– Discuss whether you can exclude the alternative interpretation that myelination is particularly prominent as the borders of certain stripes, as it as been indicated for V1 between patches and inter-patches (Pistorio et al. 2006). This might also explain contradictory results between papers.

Thank you for the insightful recommendation pointing us to this article. This is an intriguing alternative hypothesis which we were not aware of. Due to the limited spatial resolution of our methods, we cannot exclude this possibility. We therefore included it in the Discussion section (p. 10, lines 330–337). It is tempting to argue that, since we defined stripe ROIs at different z-score threshold levels (and thus including more or less border volume), we might have seen a particular discontinuity in R1 differences if only stripe borders were more myelinated, which we did not. However, this is rather speculative and presumably out of reach considering the resolution and data quality of our study.

– A brief discussion should be included on the adequacy of all the chosen visual stimuli, specifically for(i) activating (or not) foveal representations of V2 (add spatial scale to the discussion of absence of blue cones in the fovea), especially for binocular disparity.(ii) the artefactual luminance responses that red-blue gratings can generate even at isoluminance (because of chromatic aberrations: red and blue light cannot be in focus on the retina at the same time) and whether this could have activated for instance pale stripes,and hence (iii) whether one could reasonably expect that the chosen stimuli will not activate pale stripes or whether there might be overlap (with specific reference also to the chosen binocular disparity stimulus).

Regarding point (i): In the original Discussion section, we already shortly discussed the absence of activation in the region around the foveal representation (see p. 10, lines 290–303) and its potential causes including absence of blue cones and eccentricity dependence of disparity tuning. Concerning the color contrast, it is known that blue cones are absent in the central 0.5° of the retina and the macula lutea covers the central 5° of the retina (Tootell and Nasr, 2015).

It would be insightful to mark this range of the visual field in activation maps. One possibility to demarcate this region would be to use the acquired retinotopy data and use a population receptive field (pRF) model to estimate for each voxel the visual space it is most sensitive. However, the acquired retinotopy data was primarily optimized for localization of V1/V2 and V2/V3 borders and more data would be beneficial for robust pRF modeling. Furthermore, a retinotopic response of the central fovea is difficult to achieve. Therefore, we did not attempt to use our retinotopy data to get a precise mapping (spatial scale) of the central region. However, we have used it to approximate a region including the fovea and performed the data analyses excluding this region (see above), in order to identify potential biases.

Concerning the disparity contrast, it is known that many cells are clustered in disparity zones and only respond to stimuli over a narrow range (Ts’o et al., 2001). However, it remains speculative if the covered eccentricity range in the stimulus can explain absence of activation in the foveal representation.

In this regard, it might be insightful to compare our activation maps with a previous human fMRI study, which studied cortical regions selective for stereoscopic processing (Tsao et al., 2003). The study was conducted at 3T at conventional resolution (isotropic 3.1 mm voxel size, note that this is a 58-fold larger nominal voxel volume compared to our study). Critically, this study used a very similar disparity stimulus and examined the same depth range (±0.22°). This study revealed a very similar topography (due to the larger voxel size, no disparity-selective thick stripes could be resolved) with absence of activation around the foveal representation. We now added other possible explanation for the missing of disparity-selective activation around the foveal representation in the revised manuscript (p. 10, lines 295–301).

We cannot pinpoint any of these explanations. However, since we could confirm our findings after exclusion of the V2 regions around the central fovea, we conclude that missing activation around the central fovea did not bias the results in the present study in a relevant way.

Regarding (ii): For the color experiment, we used isoluminant red/blue gratings. As pointed out by the reviewer, chromatic aberration means that different colors from the same object in the visual field stimulate the retina at different locations. Thus, blue and red stimuli may not have been well matched, compromising the fMRI subtraction design. To address this issue, our stimulus included several properties to minimize chromatic aberration: (1) We used a low spatial frequency for gratings to tap the relative higher sensitivity to color (relative to luminance) at lower spatial frequencies and minimize chromatic aberration at color borders (Tootell and Nasr, 2017). (2) Additionally, the grating had a sinusoidal shape which further decreased chromatic aberration at color borders. However, it was shown in a previous study that waveform of red/blue gratings had only a marginal influence on color-selective thin stripe activation (Tootell and Nasr, 2017).

Moreover, it is known from optical imaging that color-selective thin stripes in V2 possess a systematic topography of hue representation (Xiao et al., 2003). This means that, in principle, other hues than red/blue might be used to activate color-selective thin stripes. A recent high-resolution fMRI study in humans investigated the processing of different hues (red, yellow, green, blue) in color-selective thin stripes of V2 (Nasr and Tootell, 2018). Consistent with Xiao et al. (2003), they found that thin stripes were responsive to all tested hues. However, the response of mid-spectral hues (green, yellow) compared to end-spectral hues (red, blue) showed a more uniform response across V2. That means that in contrast to mid-spectral hues, red/blue colors are suited to selectively activate V2 thin stripes.

An impressive example is the study by Tootell et al. (2004), which used the double-label deoxyglucose technique to map thin stripes in macaques while stimulating with very similar red/blue gratings compared to the present study. Derived activation maps were compared to histological sections after staining with CO. It can be seen that color-selective activation clusters are exclusively located within CO dark thin stripes (see Figure 3 in Tootell et al., 2004). We added a statement in the methods section to underline the applicability of the used colors for the stimulation of color-selective thin stripes (p. 12, lines 419–421).

Regarding (iii): Indeed, it is expected that color-selective and disparity-selective activation clusters will overlap to some extent. As pointed out in the previous paragraph, color-selective activation is confined to CO dark stripes. However, disparity-selective clusters are found to some extent in all stripes (but mostly in dark CO thick stripes). We stated this on p. 4 (lines 125–128). Furthermore, the effective spatial resolution of the BOLD response is limited not only due to technical but also physiological mechanisms, leading to potential overlap between stripes. This is discussed on p. 10 (lines 304–320). This was one of the reasons to use different z-score threshold levels in the analysis. Furthermore, all vertices with overlapping activation from color-selective thin and disparity-selective thick stripe measurements were excluded in thin and thick stripe ROIs, to dichotomize the data set.

(2) Reconsider the easy, unnecessary straw-man of "all histological results are conflicting" in abstract and introduction for a more considered analysis of the discussed papers and of what is the most likely "ground truth" that you would expect to find with MRI. For instance, Luxol not only stains phospholipids of the neuronal myelin sheath but is also associated with staining mitochondria (Takaya, 1967), which might lead to differential results in cortical staining for CO-rich compartments. Furthermore, the contradictory results for myelin staining "of" CO compartments in the literature might also have to do with more recent carefully analysis showing that myelin might be particularly associated with borders of CO blobs and inter blobs due to their relationship with OD columns (Pistorio et al. 2006). On close inspection, some of the earlier papers (e.g. Tootle et al. 1983) present little clear data on the alignment of the different maps in the papers; even with the more recent, careful studies, one needs to take a close look at the conditions tested (e.g. monocular enucleation, Horton & Hocking 1997). The discussion (e.g. lines 183 -185) seems to be in parts more considered.

Thank you very much for these comments and the insightful article recommendations. We toned down the claims about the specificity of R1 values towards myelin throughout the manuscript. We also removed the unfortunate formulation that gave the impression that MRI can resolve the conflicting results found in histological studies. In the introduction, we changed the corresponding paragraph by emphasizing the alternative and complementary view which MRI offers by the possibility to investigate structure-function relationships in the living human brain (see p. 2, lines 68–70; p. 3, lines 93–95).

Because of the contradictory findings of histological studies, we could not further finesse the hypothesis beyond that we expect differences in the myelin sensitive MRI metrics between the thin/thick versus pale stripes. If the reviewer has a particular refined hypothesis or statement in mind, which we may have overlooked, we would be grateful for their feedback. To improve the contextual understanding, we added a paragraph in the Discussion section covering in more depth how the MRI results relate to known histological findings (see pp. 8–9, lines 216–240).

Reviewer #2 (Recommendations for the authors):The Introduction strongly gives the impression that quantitative MRI emerged in the mid-2010s, and entirely from the co-authors laboratory.

We did not intend to give the impression that we founded the research field of qMRI or that the concept is as recent as the 2010s. We changed the introduction accordingly, e.g., by stating that qMRI is a collective term of techniques (see p. 2, lines 70–76) which includes the concept of physical modeling of MRI data. At the same time, we tried to stay as general as possible to accommodate the diverse readership of *eLife*.

On p10, the statement "To detect signal changes with magnitudes of a few percent, qMRI is advantageous since it is less biased by inhomogeneities in the radiofrequency transmit and receive fields, and therefore allows better comparison of differences between ROIs and across participants" doesn't make sense to me. I gather the authors intend this to be taken as an argument for qMRI rather than non-quantitative "weighted" imaging. There are a few oddities of logic here. First, the fact that these differences are a few percent in units of R1 does not necessarily mean they are a few percent in terms of weighted contrast; second, even if the signals are of similar magnitude of percentage, the process of quantification "introduces" noise (e.g., involves propagation of error that amplifies noise from the weighted images) making it harder to detect those small percent signals; third, this statement conflates effect size (% signal) with sources of bias. I can believe that avoiding sources of bias is a major argument for qMRI, but this statement needs editing.

Thank you for this comment. For brevity, we had shortened the discussion clearly too much at this point. We now elaborate further on the differences between qMRI and weighted MRI. Since a comprehensive discussion would distract from the main points of the paper we also refer to our recent review paper on qMRI (Weiskopf et al., 2021). In the following, we comment on the points raised by the reviewer:

(1) In our study, the contrast (i.e. relative percent difference between thick/thin and pale stripes) is comparable for the R1, R2* metrics and the respective weighted gradient echo images. Since the observed relative differences in R1 and R2* are in the range of a few percent, the Ernst equation and free induction decay, will lead to relative weighted signal differences (i.e. contrast) that are also in the range of a few percent and no relevant contrast increase over qMRI parameters can be expected. For example, assuming a R1 of 0.58 s^-1^ for cortical gray matter and a ΔR1 of about 0.014 s^-1^, which corresponds to our findings (see p.7, lines 185–188), the maximum achievable relative signal change is on the order of about 2% (see Author response image 1), irrespective of the chosen TR and flip angle. This percent change is comparable to the observed change of 2.4% in R1.

**Author response image 1. sa2fig1:** Relative signal changes based on R1 differences. Based on the Ernst equation, the set of curves shows the relative signal change in percent unit between the signal at R1 and the signal at R1+ΔR1 for different repetition times (TR) and flip angles. T1-weighting in gradient echo sequences increases for small TR and large flip angles. We set the R1 = 0.58 s^-1^ (which is the mean cortical gray matter R1 of our cohort in V2) and ΔR1 = 0.014 s^-1^ (see p. 7, lines 185–188), which corresponds to a relative R1 change of about 2.4%. In the graphic, it can be seen that this change in R1 translates to a similar maximal relative signal change of ca. 2% in the weighted gradient echo images.

(2) The estimation if and how the signal-to-noise ratio (SNR) or contrast-to-noise ratio (CNR) is decreased in qMRI versus weighted imaging is a highly technical and complex issue and beyond the scope of our paper. However, we briefly summarize the main aspects in the following. Assuming constant contrast (see previous point), the different noise sources need to be distinguished, particularly thermal, physiological and other sources of instrumental noise.

We agree that the quantification of R1, which requires a non-linear operation combining two differently weighted source images, leads to a reduction of signal-to-thermal noise ratio when compared against a weighted image acquired with optimal parameters. However, at 7 T, the optimal acquisition parameters are highly location dependent, since the RF transmit field varies across the brain. Thus, it is a rather academic argument that does not translate into practice.

When other sources of noise such as differences in signal due to RF transmit/receive field differences are considered, qMRI clearly outperforms weighted imaging (see e.g. Figure 1b in Weiskopf et al., 2021). At 7 T this also applies to small regions such as V2.

Acquisition protocols are frequently different between qMRI and weighted imaging. For example, the standard weighted gradient echo sequences usually acquire one image at a single echo time. The MPM protocol acquired 6 echoes, i.e. 6x the amount of data. This significantly increases the SNR compared to the conventional approach while maintaining accuracy due to appropriate modeling such as ESTATICS (Weiskopf et al., 2014). Similar considerations apply in the case of physiological noise impact, which was significantly reduced in our study due to the use of prospective motion correction (Vačulciaková et al., 2022). We modified the statements in the Discussion section accordingly, but we hope that the reviewer agrees that a comprehensive discussion would be beyond the scope of the paper (p. 9, lines 273–285).

(3) We are not sure what the “conflates effect size (% signal) with sources of bias“ statement refers to, but we presume it may refer to the use of CNR, which takes the ratio of the effect of interest (i.e. signal/parameter difference between thick/thin and pale stripes) divided by the relevant noise metric. This metric is useful for estimating the statistical power and detection probability in a measurement and in our opinion actually the most relevant metric. Note that we also stated the coefficient of variation on p. 9 (lines 279–280). Regarding the noise metric we point out that for the purpose of a ROI analysis across a large brain area such as V2, the bias is a relevant noise source in the statistical analysis, since it cannot be reliably removed. To clarify our point, we removed this sentence and revised the paragraph (p. 9, lines 273–285). We hope that the revised phrasing could clarify this point.

On p10, the authors note that the small effect sizes "hindered a direct visualization of stripes at the voxel level in R1 maps" and refer to Figure 4. But this cannot be seen in Figure 4. I recommend adding a direct comparison of flat maps of R1 and z-score so readers can see what the data looks like. Ideally this would superimpose some of the structure from fMRI (potentially the blue dots from Figure 2, although these may obscure any structure that is present).

Thank you for pointing this out. In addition to the R1 maps on inflated surfaces in the original version of the manuscript, we have included Figure 4-Supplement 1, which shows the whole-brain topography of cortical R2* for one representative participant. Furthermore, we have included now Appendix 1, which presents functional activation maps for color- and disparity-selective stripe measurements, retinotopy data (eccentricity, polar angle), and qMRI parameters maps (R1, R2*) on flattened surfaces covering V1 and V2 for each participant and both hemispheres.

References

Haast RAM, Ivanov D, Formisano E, Uludag K. Reproducibility and Reliability of Quantitative and Weighted T1 and T2* Mapping for Myelin-Based Cortical Parcellation at 7 Tesla. Front Neuroanat 2016; 10:112.

Marques JP, Kober T, Krueger G, Van der Zwaag W, Van de Moortele P-F, Gruetter R. MP2RAGE, a self bias-field corrected sequence for improved segmentation and T1-mapping at high field. Neuroimage 2010; 49:1271–1281.

Nasr S, Polimeni JR, Tootell RBH. Interdigitated Color- and Disparity-Selective Columns within Human Visual Cortical Areas V2 and V3. J Neurosci 2016; 36:1841–1857.

Nasr S, Tootell RBH. Columnar organization of mid-spectral and end-spectral hue preferences in human visual cortex. Neuroimage 2018; 181:748–759.Petracca M, El Mendili MM, Moro M, Cocozza S, Podranski K, Fleysher L, Inglese M. Laminar analysis of the cortical T1/T2-weighted ratio at 7T. Neurol-Neuroimmunol 2020; 7:e900.

Schmidt J, Radunsky D, Scheibe P, Ben-Eliezer N, Weiskopf N, Trampel R. Quantification of transverse relaxation times in vivo at 7T field-strength. Proceedings International Society for Magnetic Resonance in Medicine 2021; Abstract #3049.

Tootell RBH, Nelissen K, Vanduffel W, Orban GA. Search for Color 'Center(s)' in Macaque Visual Cortex. Cereb 2004; 14:353–363.Tootell RBH, Nasr S. Primate Color Vision. In: Brain Mapping. An Encyclopedic Reference. Elsevier 2015; 2:489–506.

Tootell RBH, Nasr S. Columnar Segregation of Magnocellular and Parvocellular Streams in Human Extrastriate Cortex. J Neurosci 2017; 37:8014–8032.

Tsao DY, Vanduffel W, Sasaki Y, Fize D, Knutsen TA, Mandeville JB, Wald LL, Dale AM, Rosen BR, Van Essen DC, Livingstone MS, Orban GA, Tootell RBH. Stereopsis Activates V3A and Caudal Intraparietal Areas in Macaques and Humans. Neuron 2003; 39:555–568.

Ts’o DY, Roe AW, Gilbert CD. A hierarchy of the functional organization for color, form and disparity in primate visual area V2. Vis Res 2001; 41:1333–1349.

Vačulciaková L, Podranski K, Edwards LJ, Ocal D, Veale T, Fox NC, Haak R, Ehses P, Callaghan MF, Pine KJ, Weiskopf N. Combining navigator and optical prospective motion correction for high-quality 500 μm resolution quantitative multi-parameter mapping at 7T. MRM 2022; 88:787–801.

Weiskopf N, Callaghan MF, Josephs O, Lutti A, Mohammadi S. Estimating the apparent transverse relaxation time (R2*) from images with different contrasts (ESTATICS) reduces motion artifacts. Front Neurosci 2014; 8:278.

Weiskopf N, Edwards LJ, Helms G, Mohammadi S, Kirilina E. Quantitative magnetic resonance imaging of brain anatomy and in vivo histology. Nat Rev Phys 2021; 3:570–588.

Xiao Y, Wang Y, Felleman DJ. A spatially organized representation of colour in macaque cortical area V2. Nature 2003; 421:535–539.